# Replay as wavefronts and theta sequences as bump oscillations in a grid cell attractor network

Louis Kang[1,2]*, Michael R DeWeese[1,2]

[1]Redwood Center for Theoretical Neuroscience, Helen Wills Neuroscience Institute, University of California, Berkeley, Berkeley, United States; [2]Department of Physics, University of California, Berkeley, Berkeley, United States

**Abstract** Grid cells fire in sequences that represent rapid trajectories in space. During locomotion, theta sequences encode sweeps in position starting slightly behind the animal and ending ahead of it. During quiescence and slow wave sleep, bouts of synchronized activity represent long trajectories called replays, which are well-established in place cells and have been recently reported in grid cells. Theta sequences and replay are hypothesized to facilitate many cognitive functions, but their underlying mechanisms are unknown. One mechanism proposed for grid cell formation is the continuous attractor network. We demonstrate that this established architecture naturally produces theta sequences and replay as distinct consequences of modulating external input. Driving inhibitory interneurons at the theta frequency causes attractor bumps to oscillate in speed and size, which gives rise to theta sequences and phase precession, respectively. Decreasing input drive to all neurons produces traveling wavefronts of activity that are decoded as replays.

*For correspondence:
louis.kang@berkeley.edu

**Competing interests:** The authors declare that no competing interests exist.

## Introduction

The hippocampal region contains spatially tuned cells that generally encode an animal's position in its spatial environment. Place cells in the hippocampal formation fire at one or a few locations (*O'Keefe and Dostrovsky, 1971*), and grid cells in the medial entorhinal cortex (MEC) fire at many locations that form a triangular lattice in space (*Hafting et al., 2005*). However, at short timescales, these neurons fire in coordinated sequences that represent trajectories away from the current animal position.

The first type of sequence occurs when the animal is moving and the local field potential (LFP) of the hippocampal region is dominated by an oscillation of 5–11 Hz, the theta range (*Vanderwolf, 1969*; *O'Keefe, 1976*; *Vinogradova, 1995*). During every cycle of this oscillation, neurons corresponding to locations slightly behind the animal fire first, followed by those corresponding to the current location and finally locations ahead of the animal (*Skaggs et al., 1996*; *O'Neill et al., 2017*). These so-called theta sequences involving many neurons are related to a single-neuron phenomenon called phase precession (*O'Keefe and Recce, 1993*; *Hafting et al., 2008*). When an animal first enters the firing field of a place or grid cell, the neuron spikes late in the theta cycle. As the animal moves through the field, subsequent spikes tend to arrive at smaller theta phase, or earlier within a cycle. Thus, activity within each theta cycle starts with neurons whose peak firing occurs behind the animal and ends with neurons whose peak firing occurs ahead of the animal, which is a theta sequence (*Skaggs et al., 1996*).

The second type of sequence occurs when the animal is idle and not moving, a state called quiet wakefulness or quiescence; it also occurs during slow wave sleep, which will not concern us here. During idle periods, the LFP of the hippocampal region loses theta oscillations but is instead

intermittently punctuated by sharp-wave ripples with power across 140–200 Hz (*Buzsáki, 1986*; *Chrobak and Buzsáki, 1996*). During these events, spatially tuned neurons fire in coordinated bursts that represent rapid, long-distance trajectories. These replays are well-established in place cells (*Lee and Wilson, 2002*; *Foster and Wilson, 2006*) and have recently been observed in grid cells within superficial layers of the MEC (*O'Neill et al., 2017*). Note that replays can also involve grid cells within *deep* layers of the MEC (*Ólafsdóttir et al., 2016*); these findings, which have been disputed (*Trimper et al., 2017*), will not be addressed by this work.

Hippocampal experiments on theta sequences and replay have established a rich phenomenology, which includes their properties near branch points of a maze (*Wu and Foster, 2014*; *Belchior et al., 2014*), their modulation by reward history (*Johnson and Redish, 2007*; *Wikenheiser and Redish, 2015a*; *Ambrose et al., 2016*; *Wu et al., 2017*), their ability to predict future actions (*Pfeiffer and Foster, 2013*; *Singer et al., 2013*; *Wikenheiser and Redish, 2015b*; *Xu et al., 2019*), and their potential role in disease pathogenesis (*Middleton et al., 2018*). These findings suggest that hippocampal sequences facilitate many cognitive functions, such as memory consolidation, credit assignment, and action planning (*Foster, 2017*; *Ólafsdóttir et al., 2018*; *Zielinski et al., 2018*). However, the mechanisms that produce theta sequences and replay among place and grid cells are still unclear.

Meanwhile, a mechanism for producing grid cells themselves, the continuous attractor model, has been accumulating experimental support (*Couey et al., 2013*; *Yoon et al., 2013*; *Heys et al., 2014*; *Dunn et al., 2015*; *Fuchs et al., 2016*; *Zutshi et al., 2018*; *Gu et al., 2018*). This class of networks posits particular configurations of synapses within the MEC that are local and symmetric (*Burak and Fiete, 2009*; *Widloski and Fiete, 2014*). By incorporating key biological features such as fully spiking neural dynamics, we find that new phenomena emerge in this recurrent architecture: grid cells will exhibit either theta sequences or replays depending on the external input provided by other brain regions. These inputs correspond to experimentally measured changes between active and inactive behavior (*Steriade et al., 2001*; *Simon et al., 2006*) and do not involve changes in connectivity. From this perspective, the same architecture that produces grid cells causes them to participate in sequences that may be leveraged by other brain regions, such as the hippocampus, to pursue desired cognitive goals.

## Results

### Our fully spiking continuous attractor model produces grid cells

We first describe our implementation of a continuous attractor network and demonstrate that it reproduces classic grid responses when a simulated animal explores a 2D open field. Model details are provided in Materials and methods and *Table 1*. We simulate a single grid module—that is, grid cells of one scale and orientation (*Stensola et al., 2012*)—with a continuous attractor network based on *Burak and Fiete (2009)* and *Widloski and Fiete (2014)*. Neurons are assembled in a 2D sheet with five overlapping populations, of which four represent excitatory principal cells and one represents inhibitory interneurons (*Figure 1A*). Each excitatory neuron excites its neighbors across all populations, and each inhibitory neuron inhibits only excitatory neurons located a certain distance away. We will address the purpose of having four slightly different excitatory populations shortly.

To accurately simulate rapid sequences of spikes, we implement fully spiking grid cells that obey leaky integrate-and-fire dynamics. Each neuron has a membrane potential that tends toward a steady-state value called the drive, which represents a combination of resting potential and broad input from other brain regions (*Figure 1B*). If the potential exceeds a threshold of 1 in arbitrary units, a spike is emitted, the potential is reset to 0, and the neuron's targets experience a postsynaptic jump in potential after a brief synaptic delay. During locomotion, excitatory principal cells are driven to fire by various cortical and subcortical inputs (*Kerr et al., 2007*; *Bonnevie et al., 2013*). Thus, their drive exceeds threshold at the center of the neural sheet; it decays towards the edges to avoid edge effects that disrupt continuous attractor dynamics (*Burak and Fiete, 2009*). Meanwhile, inhibitory interneurons have subthreshold drive that oscillates at a theta frequency of 8 Hz due to input from the medial septum (*Brandon et al., 2011*; *Koenig et al., 2011*; *Gonzalez-Sulser et al., 2014*; *Unal et al., 2015*). With this architecture and random initial membrane potentials, the neural sheet naturally develops local regions of activity, called bumps, that are arranged in a triangular lattice and

**Table 1.** Main model parameters and their values unless otherwise noted.
Values that change between runs, idle periods, and allocentric corrections are indicated accordingly.

| Parameter | Variable | Value |
|---|---|---|
| Neurons per population | $n \times n$ | $232 \times 232$ |
| Simulation timestep | $\Delta t$ | 1 ms |
| Exc. membrane time constant | $\tau_m^+$ | 40 ms |
| Inh. membrane time constant | $\tau_m^-$ | 20 ms |
| Exc.-to-exc. synaptic delay | $\tau_s^{++}$ | 5 ms |
| Exc.-to-inh. synaptic delay | $\tau_s^{-+}$ | 2 ms |
| Inh. synaptic delay | $\tau_s^-$ | 2 ms |
| Exc. drive maximum | $a_{max}^+$ | $\begin{cases} 2.0 & runs \\ 1.6 & idle \\ 2.0 & allo. \end{cases}$ |
| Exc. drive minimum | $a_{min}^+$ | 0.8 |
| Exc. drive scaled spread | $\rho_{a^+}$ | $\begin{cases} 1.2 & runs \\ 0.9 & idle \end{cases}$ |
| Inh. drive magnitude | $a_{mag}^-$ | $\begin{cases} 0.72 & runs \\ 0.0 & idle \\ 0.72 & allo. \end{cases}$ |
| Inh. drive theta amplitude | $a_{th}^-$ | $\begin{cases} 0.2 & runs \\ 0 & idle \\ 0 & allo. \end{cases}$ |
| Inh. drive theta frequency | $f$ | 8 Hz |
| Exc. synaptic strength | $w_{mag}^+$ | 0.2 |
| Exc. synaptic spread | $r_{w^+}$ | 6 |
| Inh. synaptic strength | $w_{mag}^-$ | 2.8 |
| Inh. synaptic distance | $r_{w^-}$ | 12 |
| Exc. synaptic shift | $\xi$ | 3 |
| Exc. velocity gain | $\alpha$ | 0.25 s/m |
| Exc. noise magnitude | $\mathrm{Var}[\zeta^{\mathrm{P}}(\mathbf{r}, t)]$ | $0.002^2$ |
| Inh. noise magnitude | $\mathrm{Var}[\zeta^-(\mathbf{r}, t)]$ | $0.002^2$ |

are coherent across populations (*Figure 1C* and *Figure 1—video 1*). This self-organized grid is an attractor state of the network.

How do we produce neurons with grid-like spatial tuning from the grid-like activity pattern on the neural sheet? This transformation is performed by the four excitatory populations, each of which corresponds to a perpendicular direction along the neural sheet and a perpendicular direction in the spatial environment. Excitatory neurons have output synapses biased in their preferred sheet direction (*Figure 1A*), and they receive more input when the animal moves along their preferred spatial direction. When the animal moves along, say, the 'North' direction, neurons in the N population have increased drive; since their outputs are biased in the 'up' direction on the neural sheet, the activity pattern moves up. In this way, the activity pattern on the sheet moves synchronously with the animal's 2D trajectory (*Figure 1D*), and the grid pattern is projected onto single neuron spatial responses (*Figure 1E* and *Figure 1—video 2*). Each active neuron in the sheet is a grid cell with the same scale and orientation (*Figure 1—figure supplement 1*).

## Grid cells are spatially tuned and theta-modulated along a linear track

For the rest of the paper, we simulate 1D trajectories consisting of runs along a linear track separated by idle periods at either end (*Figure 2A*). As in the 2D case, attractor bumps on the neural sheet move synchronously with animal motion. But over the course of a simulation, the grid-like pattern on the neural sheet drifts (*Burak and Fiete, 2009*) such that for a given track position, the

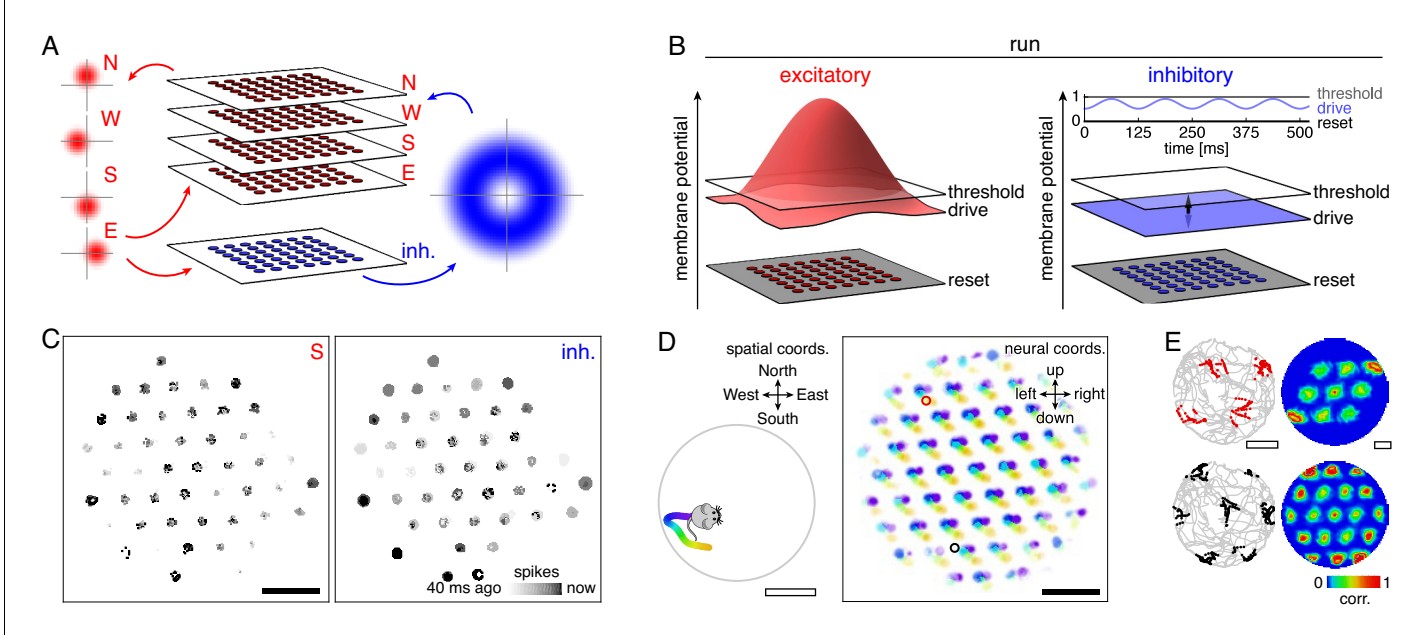

**Figure 1.** Model architecture and generation of 2D grid cells. (**A**) Our model consists of a neural sheet with five overlapping populations, four of them excitatory—N, W, S, and E—and one inhibitory. Each density plot depicts the synaptic outputs in the sheet of a neuron at the origin. (**B**) Each neuron is driven to a particular membrane potential, which exceeds the spiking threshold for excitatory neurons at the center of the sheet and oscillates at 8 Hz for inhibitory neurons while the animal is running. (**C**) Snapshot of neural activity showing S and inhibitory populations separately; other excitatory populations have activity patterns similar to that of the S population. Each pixel is a neuron and dark colors indicate recent spiking. (**D**) Left, segment of a 2D open field trajectory. Right, neural activity over the course of the segment with each neuron colored according to the position at which it attained its maximum firing rate. Each attractor bump moves in synchrony with animal motion. (**E**) Left, two sample grid cells with spikes shown as colored dots superimposed on the animal's trajectory. Each neuron's location in the sheet is indicated by a circle of corresponding color in D. Right, autocorrelation of rate maps calculated from spikes at left. Black scale bars, 50 neurons. White scale bars, 50 cm.

The online version of this article includes the following video and figure supplement(s) for figure 1:

**Figure supplement 1.** 2D firing maps for many grid cells.

**Figure 1—video 1.** Emergence of a grid-like pattern on the neural sheet from randomly initialized membrane potentials.

https://elifesciences.org/articles/46351#fig1video1

**Figure 1—video 2.** Path integration over an open field trajectory produces 2D grid cells.

https://elifesciences.org/articles/46351#fig1video2

corresponding bump locations on the neural sheet slowly change. This drift introduces errors in path-integration (*Hardcastle et al., 2015*), which are believed to be corrected by allocentric input from border cells (*Ocko et al., 2018*; *Keinath et al., 2018*), boundary vector cells (*Evans et al., 2016*), or landmark cells (*Raudies and Hasselmo, 2015*; *Savelli et al., 2017*). Thus, we implement brief allocentric corrections in our model. Attractor bump locations on the neural sheet that correspond to either end of the track are learned during simulation setup. They are periodically reintroduced during the main simulation as excitatory input between idle periods and runs (*Figure 2A,B*; see Materials and methods). In Appendix 2, we present a version of our model without allocentric

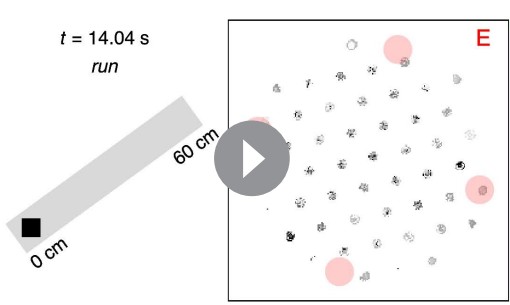

**Video 1.** Neural activity during runs, idle periods, and allocentric corrections. Left, position of the animal (black square) along a 1D track. Right, neural activity of the E population. Each pixel is a neuron, with black corresponding to current spikes and lightest gray corresponding to spikes 40 ms ago. Red circles indicate regions of recording.

https://elifesciences.org/articles/46351#video1

corrections that still demonstrates many of our results, though to a weaker degree.

Grid cells at different locations in the sheet generally correspond to different positions along the track (*Figure 2C*). From four regions in the sheet chosen for recording, we select up to 150 excitatory neurons whose spatial firing fields are stable from lap to lap and collectively tile the track (*Figure 2D*; Materials and methods). Over replicate simulations with different random initial membrane potentials, many simulations contain only grid cells with single fields as in *Figure 2D*. In other replicates, the attractor grid self-organizes on the neural sheet with an orientation such that two attractor bumps pass over recorded neurons and produce two fields (*Figure 2—figure supplement 1*; Appendix 1). All of our single-simulation examples presented below in the main text correspond to the simulation depicted in *Figure 2D* that exhibits single fields. Similarly, the grid cells in *O'Neill et al. (2017)*, with which we will primarily compare our results, often possess a single dominant field along an arm of a T-maze. However, both types of simulations contribute to our results, and we will provide examples for simulations exhibiting two fields in our figure supplements.

At the theta timescale, grid cell activity is highest when the oscillating inhibitory drive is lowest and excitatory cells are most disinhibited (*Figure 2E*). This observation allows us identify theta phases in our simulations with those defined through LFP measurements. Experimentally, grid cells (*Hafting et al., 2008*) and place cells (*Buzsáki, 2002*; *Feng et al., 2015*) are most active at troughs of the theta-filtered LFP, which can be explained in terms of extracellular currents generated by neural activity (*Buzsáki et al., 2012*). *O'Neill et al. (2017)* define theta cycles to span from trough to trough of the LFP. Accordingly, we align our theta cycles to start with phase 0° at troughs of the inhibitory drive (*Figure 2E*).

## Theta phase precession arises from oscillations in attractor bump size
### Phase precession results

During runs, activity passes through recorded neurons as attractor bumps move along the neural sheet (*Figure 3A*, *Figure 3—figure supplement 1*, and *Video 1*). However, spike trains from single neurons exhibit finer temporal organization with respect to the theta oscillation in a variety of ways (*Figure 3B*, *Figure 3—figure supplement 2*, and *Figure 3—figure supplement 3*). Many neurons are phase-independent and fire throughout the theta cycle. Some are phase-locking and strongly prefer to fire at 360° (equivalently, 0° or 720°). Finally, some exhibit phase precession with a preferred phase that starts around 360° when the animal enters a grid field but decreases as the animal progresses through it.

We first quantify these relationships using circular-linear correlation (*Equation 13*), which indicates whether theta phase changes with position (*Kempter et al., 2012*). Our correlation magnitudes 0.17 ± 0.11 (mean ± s.d.; *Figure 3C*) are low compared to experiments, which report grid cells with mean ≈ 0.3 and place cells with mean ≈ 0.4 (*O'Neill et al., 2017*). This difference arises from two sources. Figure S10 of *O'Neill et al. (2017)* shows some highly correlated neurons with magnitudes up to 0.8; these are absent from our simulations. It also shows fewer neurons with correlation value close to 0, which corresponds to either phase-independent or phase-locking neurons. Since both subgroups lack a preferred theta phase that consistently changes with position, circular-linear correlation cannot distinguish between them (*Figure 3B*, top two rows).

To further characterize phase behavior, we use circular-linear regression (*Kempter et al., 2012*), which can differentiate between all three subgroups of phase relationships presented in *Figure 3B*. Phase-independent neurons are defined to have a regression fit score less than a cutoff of 0.4 (*Figure 3D*; *Equation 11*). Neurons with high fit score are then distinguished as either phase-locking or precessing by the absolute value of their regression slope in units of field size, which is called the phase precession range (*Figure 3E*). We use a range of 60° as the cutoff. Experimentally, phase-locking and precessing neurons are found in Layers III and II (LIII and LII), respectively (*Hafting et al., 2008*). A wide variety of phase precession ranges are reported across subtypes of LII neurons with medians spanning from 50° for pyramidal cells (*Ebbesen et al., 2016*) to 170° for stellate cells (*Reifenstein et al., 2016*). Our cutoff is close to the lower limit of this span. Yet, regardless of the cutoff value, regression slopes are biased towards negative values for rightward laps and positive values for leftward laps, which means that the preferred firing phase tends to decrease as an animal moves through a grid field. This directional tendency is a key biological feature of phase precession

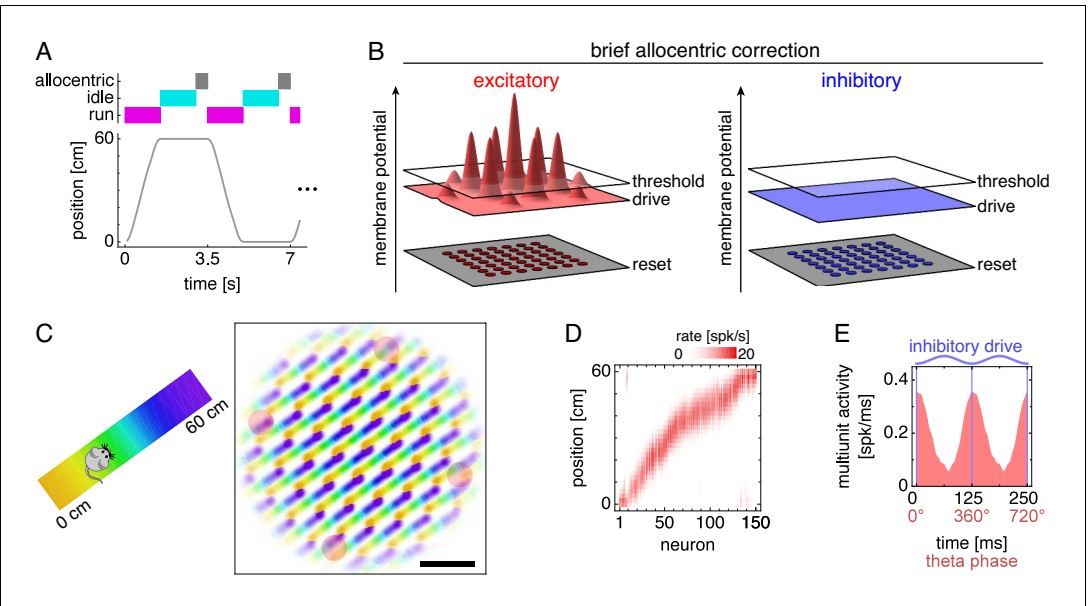

**Figure 2.** Grid cells along a 1D track. (**A**) Trajectory consisting of runs along a track separated by idle periods at either end. Between the end of an idle period and the start of a run, the network receives brief allocentric input. (**B**) Allocentric input corrects the location of attractor bumps on the neural sheet (Materials and methods). (**C**) Left, track diagram. Right, neural activity over runs with each neuron colored according to the track position at which it attained maximum firing rate. Red circles indicate regions of recording. Scale bar, 50 neurons. (**D**) Firing fields of recorded grid cells sorted by position of maximum rate. (**E**) Multiunit activity of neurons in D averaged over theta cycles, which span from one trough of the oscillating inhibitory drive to the next. Data is repeated over two cycles for clarity.

The online version of this article includes the following figure supplement(s) for figure 2:

**Figure supplement 1.** Firing fields for multiple simulations.

seen experimentally (*Hafting et al., 2008*), and it is also maintained for other cutoff values applied to the fit score (*Figure 3–Figure Supplement 4A,B*).

We also construct spike density plots using all neurons in each subgroup (*Figure 3F*). These plots show that phase-locking neurons have a preferred phase around 360° throughout the field and that phase-precessing neurons have a preferred phase also around 360° at field entry. This value corresponds to theta troughs when excitatory neurons are most disinhibited (*Figure 2E*) and matches measurements for LIII and LII neurons (*Hafting et al., 2008*; note that their phase convention differs from ours by 180°). More detailed features also seem to agree between phase-precessing neurons in our simulations and LII neurons in experiments. First, the peak density decreases by ≈75° over the first 50% of field progress, and second, there is a smaller density at around 60% field progress and 140° theta phase (*Figure 3F*; *Hafting et al., 2008*). This second density has been separately related to bursting (*Climer et al., 2013*) and a second source of neural input (*Yamaguchi et al., 2002*; *Chance, 2012*); further investigation is required to reconcile these assertions with its origin in our model.

The dependence of phase behavior on simulation features and parameters is explored in detail in the figure supplements of *Figure 3*. We will briefly list a few key results interspersed with experimental findings that support them. Phase precession statistics do not depend on the direction of animal motion (*Figure 3–Figure Supplement 5E,F*); in experiments, *Climer et al. (2013)* report omnidirectional precession in a 2D environment. Precession range and correlation magnitude increase with animal speed (*Figure 3–Figure Supplement 6A*, $p = 0.002$ and 0.0006 for precession range and correlation magnitude, respectively, by pooling data across track orientations and applying the Krushal-Wallis $H$ test); in experiments, *Jeewajee et al. (2014)* report that faster motion is associated with steeper precession slopes and higher correlation magnitudes (*Figure 3a*, results for *pdcd*, which corresponds to field progress defined in this paper).

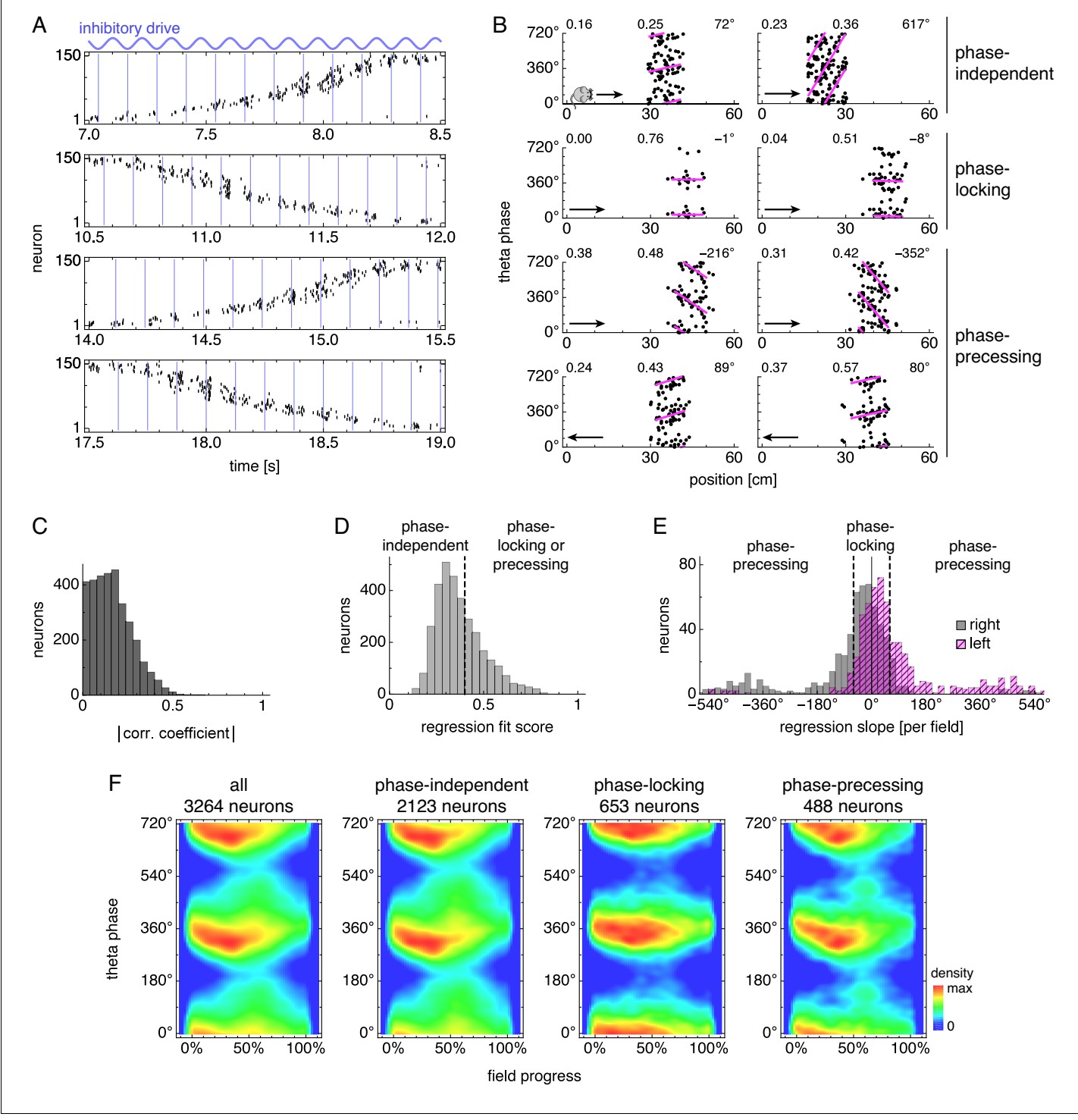

**Figure 3.** During runs, certain neurons exhibit theta phase locking and precession. (**A**) Grid cell spike rasters for four laps along the track. Vertical blue lines indicate theta cycle boundaries. (**B**) Relationship between animal position along the track and theta phase for representative neurons. Dots represent spikes during runs in the directions indicated by arrows, with each spike repeated at two equivalent phases for clarity. Lines indicate fit by circular-linear regression. Numbers in each panel from top left to top right indicate magnitudes of correlation coefficients, regression fit scores, and regression slopes. (**C–E**) Data across all replicate simulations. (**C**) Magnitudes of circular-linear correlation coefficients. Mean ± s.d.: 0.17 ± 0.11. (**D**) Fit scores for circular-linear regression. (**E**) Regression slopes in units of field size for neurons with fit score >0.4. The predominance of negative values for rightward runs and positive for leftward runs indicates decreasing phase as the animal traverses grid fields in either direction. (**F**) Spike densities for different subgroups. An animal enters a grid field at progress 0% and exits it at 100%.

*Figure 3 continued on next page*

*Figure 3 continued*

The online version of this article includes the following figure supplement(s) for figure 3:

**Figure supplement 1.** Spike rasters over multiple runs.
**Figure supplement 2.** Position-phase relationships during rightward runs for many neurons.
**Figure supplement 3.** Position-phase relationships during leftward runs for many neurons.
**Figure supplement 4.** Further analysis of subgroups with different phase relationships.
**Figure supplement 5.** Dependence of phase relationships on excitatory population and number of fields.
**Figure supplement 6.** Phase relationship properties over different simulation parameters.
**Figure supplement 7.** Phase relationship properties over different simulation parameters, continued.

To summarize our phase precession results, our simulation produces a subgroup of phase-locking neurons akin to MEC LIII neurons and a subgroup of phase-precessing neurons akin to MEC LII neurons. Further consideration of correlation coefficients and precession ranges, with connections to measurements, will be provided in the Discussion.

## Attractor bump oscillations explain phase precession

We can understand the emergence of phase precession through the effect of oscillating inhibitory drive on attractor bumps. Higher inhibitory drive suppresses the activity of the excitatory grid cells and decreases the size of bumps (*Figure 4A,B*). Imagine we record from a single neuron as a bump of activity moves through it on the neural sheet. As an attractor bump moves, its size oscillates with inhibitory drive (*Figure 4C*, top, and *Figure 4—video 1*). The activity bump will most likely first reach the grid cell when it is large and growing in size, which corresponds to the end of the theta cycle when the inhibitory drive is decreasing towards its lowest point. Thus, as an animal enters a grid field, the first spike will likely have large theta phase slightly less than 360˚. During the next theta cycle, the center of the bump has moved closer to the neuron, so the edge of the bump will reach the neuron when it is growing in size but not as large. Thus, the next spike will have slightly smaller theta phase. This is the origin of phase precession in our model. We continue generating spikes with simplified dynamics in which the neuron fires when it is contained within the bump, but there is a 40 ms refractory period corresponding to membrane potential building up towards threshold after it is reset. This procedure represents one lap along the track at constant velocity.

We represent additional laps by performing the same simplified procedure as above, but with different shifts in theta phase, since the animal does not always enter the grid field at the same initial phase. Spikes accumulated across laps show phase precession (*Figure 4C*, bottom), though the effect is strongest at the beginning of the field. In the middle of the field, spikes still precess, but they cluster around values of both 360˚ and 180˚. At the end of the field, a few spikes even precess in the opposite direction, increasing in phase with time. Thus, bump oscillations alone can explain the direction of phase precession and the preferred phase at field entry, but spike phases do not decrease perfectly linearly with progress through the field.

To analyze attractor bump oscillations more precisely, we calculate the average bump shape as a function of theta phase (*Figure 4D* and *Figure 4—video 2*; Appendix 1). We now imagine that this average bump passes multiple times through a recorded neuron whose spiking probability is proportional to bump activity (*Figure 4E*). Each instance of this stochastic process produces a phase-locking, phase-precessing, or phase-independent neuron. By explicitly rescaling the activity of the average bump without otherwise changing its time-varying shape, we can generate neurons enriched in one of these phase relationships (*Figure 4F* and *Figure 4—figure supplement 1*). Under high activity, neurons are driven to fire across all theta phases and exhibit phase independence. Under low activity, neurons only fire during the most permissive theta phase and exhibit phase locking at 360˚. Under intermediate levels of activity, neurons can respond to oscillations in bump size as demonstrated in *Figure 4C* and exhibit phase precession.

This finding on how activity level influences phase behavior is supported by the experimental report that higher spike counts are associated with steeper precession slopes (*Jeewajee et al., 2014*, *Figure 3b*, results for *pdcd*, which corresponds to field progress defined in this paper). Our main model also reflects this finding: neurons with low, medium, and high firing rates have increased likelihoods for phase locking, precession, and independence (*Figure 3–Figure Supplement 4D*).

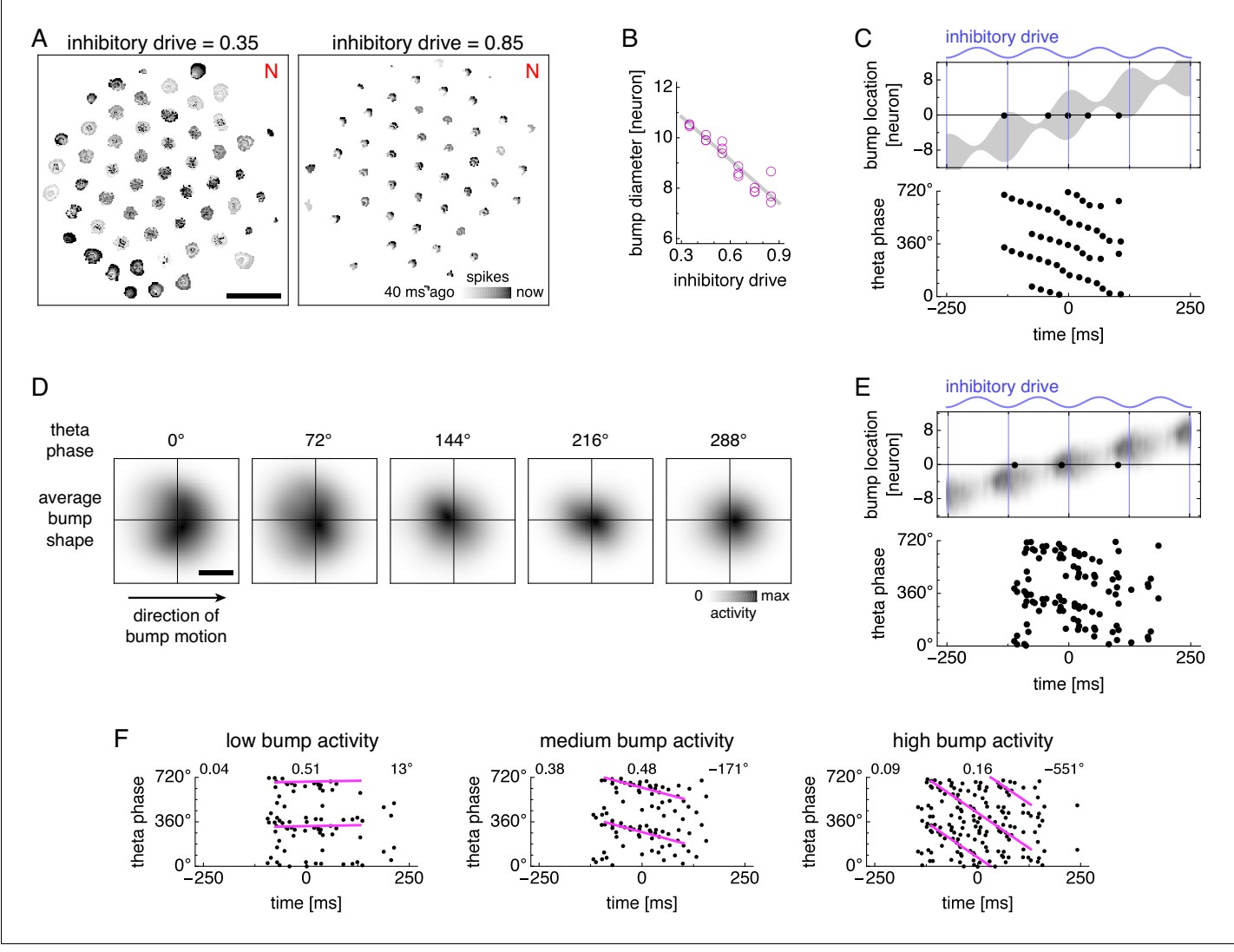

**Figure 4.** Bump size oscillations explain phase precession in models with simplified dynamics. (**A,B**) Data from simulations with fixed inhibitory drive and constant animal velocity. (**A**) Snapshots of neural activity. Scale bar, 50 neurons. (**B**) The diameter of bumps on the neural sheet decreases linearly with inhibitory drive (linear regression $R^2 = 0.90$, ANOVA $p \sim 10^{-9}$). (**C**) Phase precession in a conceptual model with bump size oscillations. We imagine an attractor bump, with size oscillations described by B, passing through a recorded grid cell. Top, a single lap. The recorded neuron is at location 0 and fires a spike (black dot) whenever contained within the bump (gray area), subject to a 40 ms refractory period. Bottom, relationship between theta phase and time across multiple laps with different initial phases. Spikes occur around 360° (equivalently, 0°) at the start of the field, and their phase generally decreases with time within in the grid field. (**D**) Attractor bump shape at different theta phases averaged over theta cycles and individual bumps (Appendix 1). Grays scaled separately for each theta phase. Scale bar, three neurons. (**E,F**) Phase behavior in a simplified model using average bump dynamics. (**E**) We imagine the average attractor bump passing through a recorded grid cell. Top, a single lap. The recorded neuron is at location 0 and stochastically fires a spike (black dot) with rate proportional to bump activity, subject to a 40 ms refractory period. Bottom, relationship between time and theta phase across multiple laps with different initial phases. (**F**) Relationship between time and theta phase using average bumps whose activity has been rescaled to different maximum values: 40, 50, and 100 spikes/s. Dots represent spikes generated according to E. Lines indicate fit by circular-linear regression. Numbers in each panel from top left to top right indicate magnitude of correlation coefficient, regression fit score, and regression slope.

The online version of this article includes the following video and figure supplement(s) for figure 4:

**Figure supplement 1.** Simplified model using average bump dynamics produces different phase behaviors for different levels of rescaled activity.

**Figure 4—video 1.** Phase precession in a simplified conceptual model.

https://elifesciences.org/articles/46351#fig4video1

**Figure 4—video 2.** Average attractor bump shape as a function of theta phase.

https://elifesciences.org/articles/46351#fig4video2

Thus, the heterogeneous phase behaviors observed in our main simulations arise, at least in part, from heterogeneous levels of bump activity. Three factors that determine the level of activity experienced by a neuron are its distance to the center of the neural sheet, its sheet location relative to attractor bumps, and its preferred firing direction relative to animal motion. Accordingly, the prevalence of different phase behaviors varies with these factors (*Figure 3–Figure Supplement 4E,G* and *Figure 3–Figure Supplement 5B,D*).

## Theta sequences arise from oscillations in attractor bump speed

Next, we use the firing fields illustrated in *Figure 2D* to decode the animal's position from the population activity presented in *Figure 3A* (Materials and methods). The decoded position generally matches the animal's actual position (*Figure 5A* and *Figure 5—figure supplement 1*), but at the theta timescale, there are deviations whose regularities are revealed by averaging over theta cycles.

To do so, we take quadruplets of consecutive theta cycles, ignoring those whose decoded positions are close to the ends of the track and reversing those corresponding to right-to-left motion (Materials and methods). We align the decoded positions with respect to the actual position of the animal midway through each theta quadruplet, and we average these shifted decoded positions over theta quadruplets. This average decoded trajectory generally increases across the quadruplet, corresponding to the animal's actual forward motion (*Figure 5B* and *Figure 5—figure supplement 2*). However, within each theta cycle, the decoded position increases rapidly, before retreating at cycle boundaries. We identify these forward sweeps as theta sequences, which represent motion at $0.88 \pm 0.15$ m/s, approximately twice the actual speed of $0.50 \pm 0.05$ m/s (mean $\pm$ s.d.; *Figure 5C*).

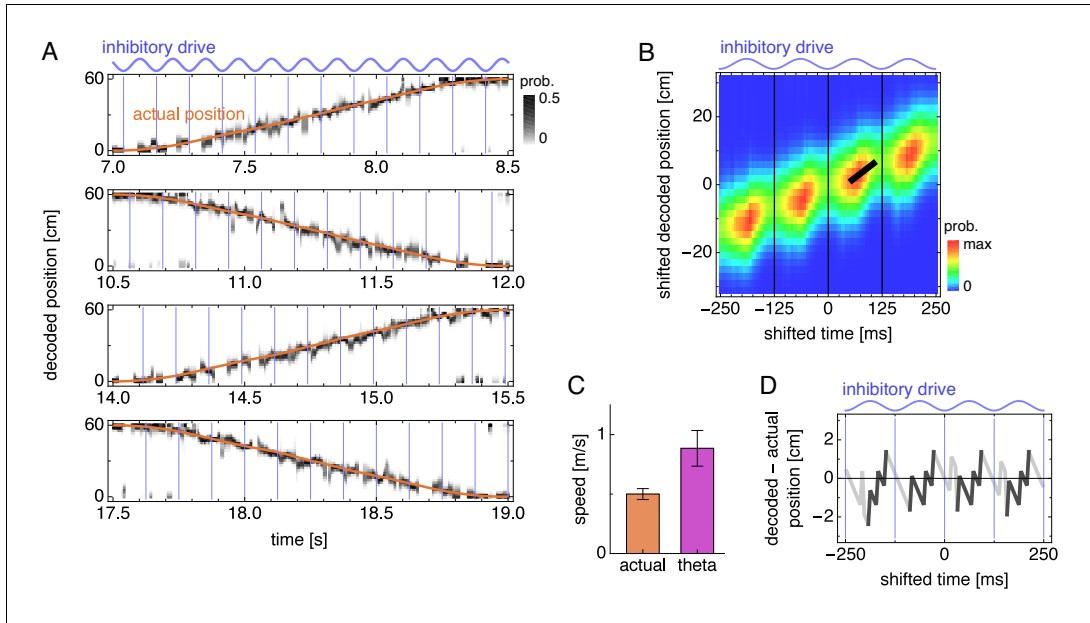

**Figure 5.** During runs, the decoded position exhibits theta sequences. (**A**) Decoded position for four laps along the track corresponding to *Figure 3A*. Vertical blue lines indicate theta cycle boundaries. (**B**) Decoded position shifted by actual position and averaged over theta cycles. The thick black fit line indicates the time window with the steepest increase in decoded position. Vertical thin black lines indicate theta cycle boundaries. (**C**) Mean actual run speed (error bars indicate s.d. over time) and mean theta sequence speed, as indicated by the slope of the thick black line in B (error bars indicate s.d. over replicate simulations). (**D**) Difference between the maximum likelihood decoded position from B and the actual animal position as a function of time. Within each theta cycle, the segment from the smallest to the largest value is emphasized as a theta sequence.

The online version of this article includes the following figure supplement(s) for figure 5:

**Figure supplement 1.** Decoded position over many runs.
**Figure supplement 2.** Decoded position shifted by actual position and averaged over theta cycles for many simulations.
**Figure supplement 3.** Dependence of theta sequence speed on number of fields and simulation parameters.

This matches experimental observations of average theta sequence speed ≈ 1 m/s compared to the animal's speed of ≈ 0.4–0.5 m/s (*O'Neill et al., 2017*). We also illustrate these sequences by comparing the maximum likelihood decoded position and the actual position as a function of time (*Figure 5D*). Averaged over theta cycles, the decoded position lags behind the actual position at cycle boundaries and then advances ahead of it midcycle.

Theta sequences, like phase precession, arise naturally in our model from an effect of oscillating inhibitory drive on attractor bumps. In addition to determining bump size, inhibitory drive also affects the speed at which bumps move along the neural sheet—higher inhibitory drive produces faster bump motion for a given animal speed (*Figure 6A,B*). At the middle of theta cycles, inhibitory drive is highest, so attractor bumps move quickly and represent fast animal trajectories. These theta sequences are faster than the animal's actual speed, which is approximately constant within a theta cycle and thus encoded as the average bump speed over the entire cycle. We can also see the origin of theta sequences directly from the average dynamics of attractor bumps (*Figure 4D*, *Figure 6C*, and *Figure 4—video 2*). The location of peak activity moves with a sawtooth-like pattern that, over a theta cycle, lags behind the overall motion of the bump and advances ahead of it (*Figure 6D*). These forward surges produce the sequences in decoded position (*Figure 5D*).

Our model's mechanism for theta sequences can explain why their speed is roughly twice the actual speed, as observed experimentally (*O'Neill et al., 2017*). Suppose the theta oscillation spends roughly the same amount of time at its peaks, when attractor bumps move at maximal speed, and at its troughs, when bumps do not move at all. Then, the average bump speed, which

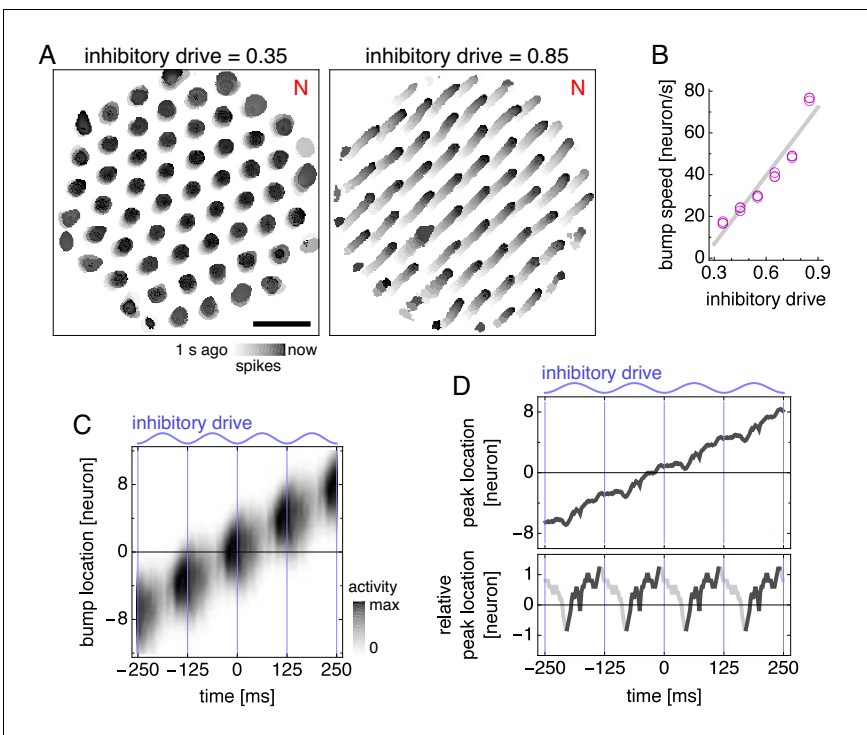

**Figure 6.** Bump speed oscillations explain theta sequences. (A,B) Data from simulations with fixed inhibitory drive and constant animal velocity. (A) Snapshots of neural activity. Scale bar, 50 neurons. (B) The speed of bumps on the neural sheet increases linearly with inhibitory drive (linear regression $R^2 = 0.91$, ANOVA $p \sim 10^{-9}$). Thus, the average decoded position in *Figure 5B* advances most rapidly around the middle of the theta cycle. (C) Activity of the average attractor bump depicted in *Figure 4D* along its central axis. (D) Top, peak activity location from C as a function of time. Bottom, peak activity position relative to constant bump motion as a function of time. Within each theta cycle, the segment from the smallest to the largest value is emphasized and corresponds to a theta sequence in *Figure 5D*.

The online version of this article includes the following figure supplement(s) for figure 6:

**Figure supplement 1.** Relationship between attractor bump size and speed.

encodes the actual animal speed, is half the maximal bump speed, which encodes the theta sequence speed. Indeed, simulations whose grid cells contain two fields, whose track is oriented differently, whose trajectory is faster or slower, or whose parameters are chosen differently all have theta sequence speeds between 1.2 and 2.5 times the actual speed (*Figure 5—figure supplement 3*).

## Replay arises from traveling wavefronts of activity

We now simulate idle periods at either end of the track. We would like to know how external inputs to the MEC change at transitions between active and quiescent states, but these measurements have not been performed to our knowledge. Thus, we instead adapt experimental findings taken during slow wave sleep, which resembles the quiescent state in many behavioral and electrophysiological ways (*Buzsáki, 1996*). The neocortex, which activates MEC principal cells, is broadly less active during slow wave sleep (*Steriade et al., 2001*), and the medial septum, which inhibits MEC interneurons, increases its firing rate while losing most of its oscillatory nature (*Simon et al., 2006*). We model these effects respectively as decreased excitatory drive and decreased inhibitory drive without oscillations (*Figure 7A*). The network rapidly switches between these inputs and those of *Figure 1B*, which represent the active state, as the animal transitions between quiescence and runs.

During quiescence, attractor bumps still form a lattice at the center of the neural sheet, where excitatory drive still exceeds threshold (*Figure 7B* and *Video 1*). However, these bumps vanish toward the edge of the sheet, where they are replaced by traveling wavefronts of activity. These wavefronts cannot be solely sustained by excitatory drive, which is subthreshold in this outer region; instead, they must be nucleated and then self-sustained through excitatory-to-excitatory connections. In our model, nucleation occurs at activity bumps toward the edge of the sheet (*Video 1*). Once nucleated, the wavefront propagates freely through the outer region because inhibitory neurons, with low drive, are not activated by the wavefront. In contrast, wavefronts cannot propagate in the center region, because excitatory neurons there receive enough drive to spike vigorously and activate nearby inhibitory neurons (*Figure 7B*, inhibitory population). These interneurons, with their surround distribution of synaptic outputs (*Figure 1A*), constrain neural activity to localized bumps and prevent wavefront propagation.

When a wavefront passes through a region of recorded neurons, it can appear as a sequence of spikes (*Figure 7C* and *Figure 7—figure supplement 1*) that is decoded as a rapid trajectory (*Figure 7D* and *Figure 7—figure supplement 2*). We identify these events as replays, which traverse much of the track over ~100 ms (*Figure 7E* and *Figure 7—figure supplement 3*). Due to the stochastic nature of wavefront generation, the number of detected replays can vary considerably across laps (*Figure 7—figure supplement 2*) and across replicate simulations (*Figure 7F*). However, replays have a characteristic speed determined by that of wavefront propagation; it is 7.0 ± 2.6 m/s, which is about 14 times faster than the actual speed of 0.50 ± 0.05 m/s (mean ± s.d.; *Figure 7G*). This agrees with experiments, which measure replays at ≈4–6 m/s and actual motion at ≈0.4–0.5 m/s (*O'Neill et al., 2017*). Replay speed depends on the number of grid fields and the values of simulation parameters (*Figure 7—figure supplement 4*). In fact, grid cells with two fields encode two different track positions, so replays among them can have their distribution of decoded position split between parallel lines (*Figure 7—figure supplement 3*). We have implemented two different detection methods that yield slightly different replay speeds: one that is identical to the single-field case, and another in which multiple lines participate in the fit (*Figure 7—figure supplement 4A-D* ; Appendix 1). Nevertheless, across all simulation parameters and detection methods, replays are always much faster than both the actual animal speed and the theta sequence speed, as observed experimentally (*O'Neill et al., 2017*). This happens because traveling wavefronts lack inhibition and should be the network phenomenon that propagates fastest.

## Replay properties under different network configurations
### Lower velocity gain produces replays with directional bias

Under the main parameter values in *Table 1*, attractor bumps travel long distances on the neural sheet when the animal moves from one end of the track to another (*Figure 2C*). These distances are longer than the grid scale, so different positions on the track can correspond to the similar locations on the neural sheet, preventing a consistent correspondence between neural sheet location and

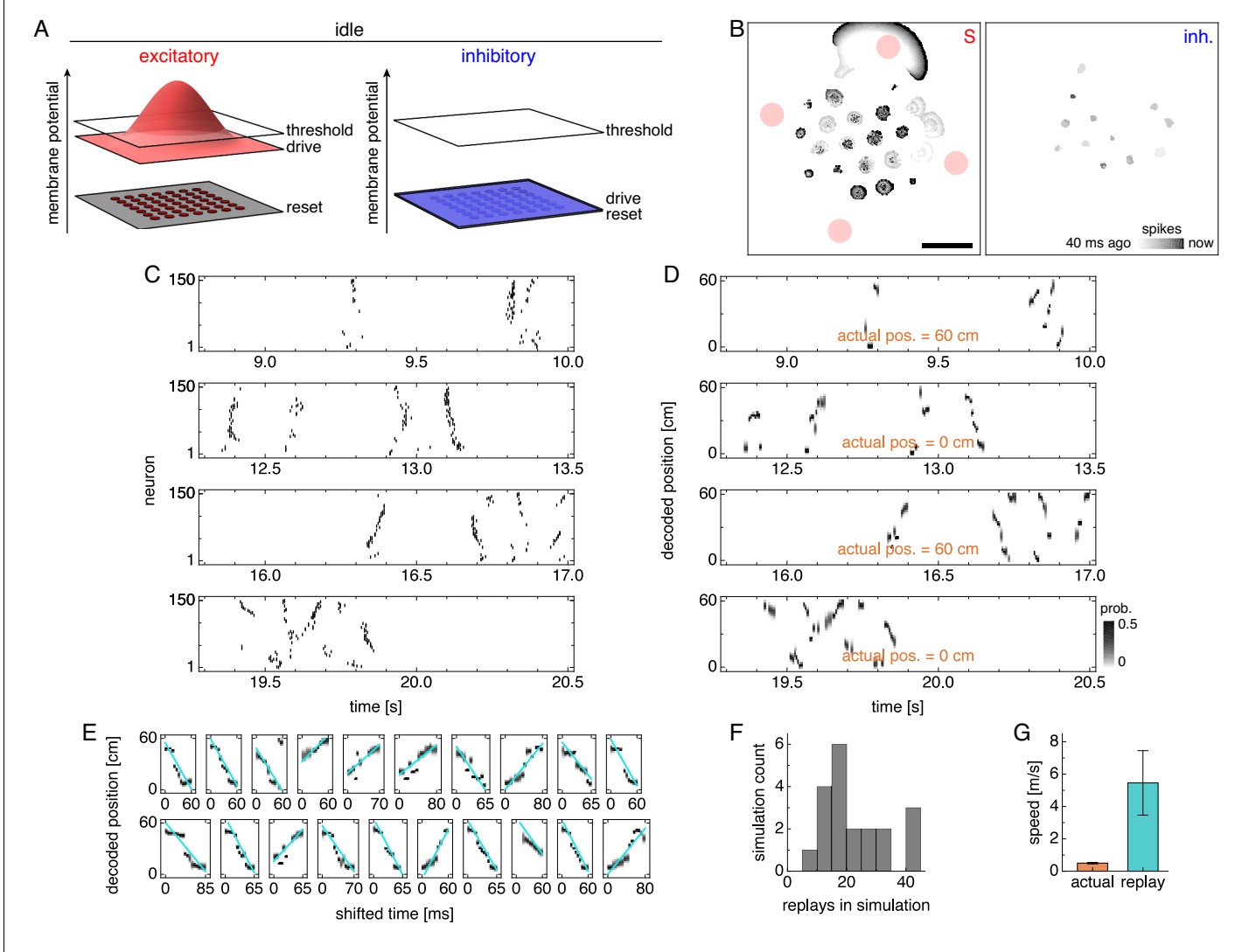

**Figure 7.** Lower drive during quiescence produces traveling wavefronts of activity that are decoded as replays. (**A**) During idle periods at either end of the track, excitatory and inhibitory drives decrease, and the latter no longer oscillates. (**B**) Snapshot of neural activity showing S and inhibitory populations separately. Red circles indicate regions of recording. Scale bar, 50 neurons. (**C**) Grid cell spike rasters for four idle periods. (**D**) Decoded position corresponding to C. Rapid trajectory sweeps arise from traveling wavefronts as depicted by the S population in B. (**E**) Replays from one simulation with cyan fit lines. (**F**) Number of replays across simulations. (**G**) Mean actual run speed (error bars indicate s.d. over time) and mean replay speed, as indicated by the slopes of cyan lines in E (error bars indicate s.d. over replays).

The online version of this article includes the following figure supplement(s) for figure 7:

**Figure supplement 1.** Spike rasters over multiple idle periods.

**Figure supplement 2.** Decoded position over many idle periods.

**Figure supplement 3.** Many replays.

**Figure supplement 4.** Dependence of replay count and speed on number of fields and simulation parameters.

track position (yellow and purple areas in *Figure 2C*). With lower velocity gain, attractor bumps travel shorter distances, and such a correspondence is apparent (yellow and purple areas in *Figure 8A*).

When an animal is idle at either end of the track, attractor bumps are still located at neurons that represent the current position (*Figure 8A* and *Figure 8—video 1*). Recall that replays are nucleated at attractor bumps (*Video 1* and *Figure 8A*). As a wavefront radiates from a bump, it will likely first activate neurons whose firing fields represent nearby positions, followed by those with increasingly distant fields, due to the correspondence between neural sheet locations and track positions

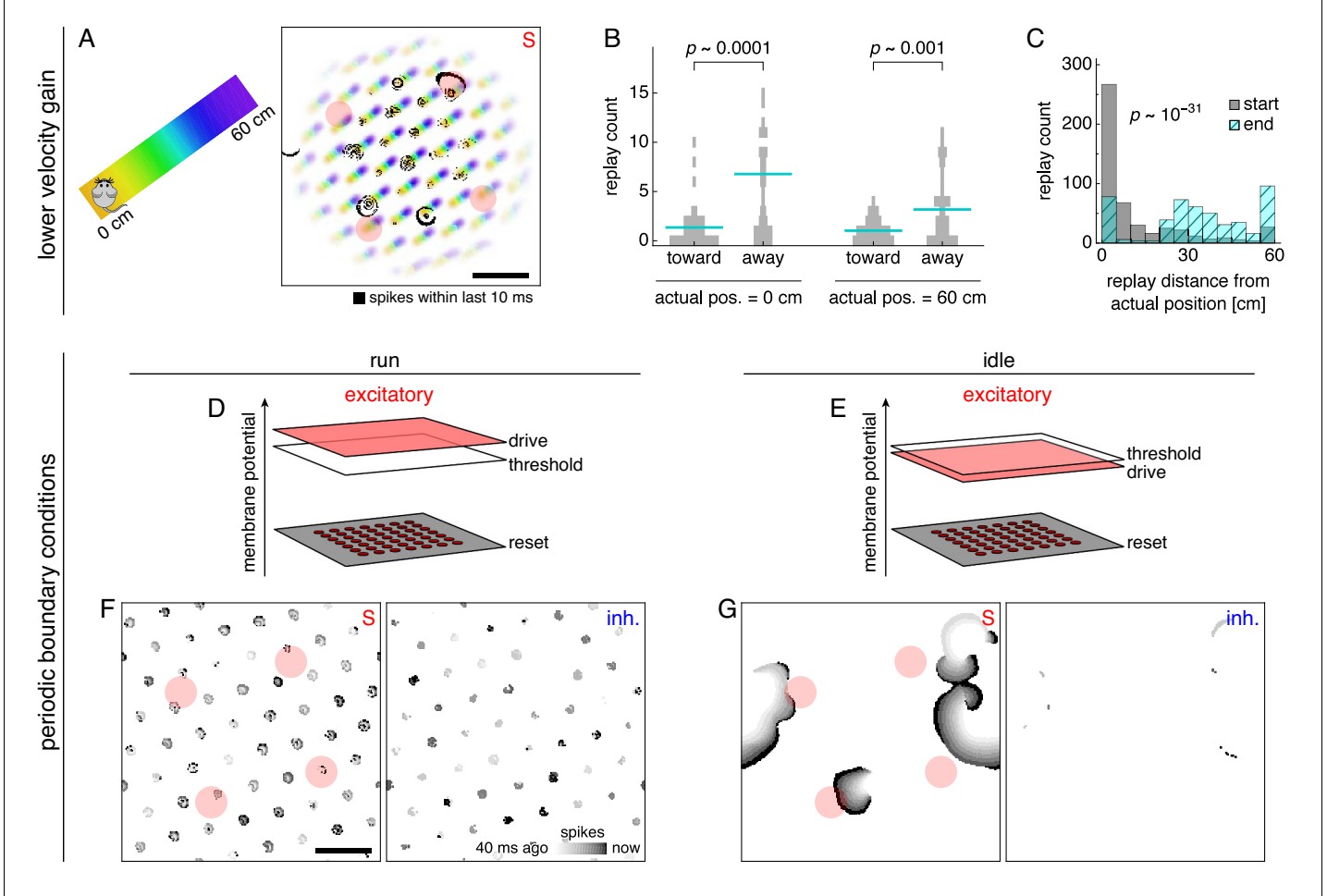

**Figure 8.** Replays acquire additional properties in networks with different configurations of attractor bumps. (A–C) With lower velocity gain, neural sheet locations have a clear one-to-one correspondence with track positions, which allows replays to exhibit a directional bias. (A) Left, track diagram with animal idle at 0 cm. Right, corresponding snapshot of neural activity superimposed on a background colored according to the position at which each neuron attains its maximum firing rate (cf. *Figure 2C*, right panel). Red circles indicate regions of recording. Attractor bumps still represent the animal position, and a wavefront can be seen emanating from one bump at the top right. (B) Direction of replay propagation across simulations. For each actual position and replay direction, width of gray bars indicates the number of simulations with a particular replay count. Means indicated with lines. *p*-values from the Mann-Whitney *U* test show that a significantly greater number of replays propagate away from either actual position. (C) Distances between actual position and decoded position at the start or end of replays. Paired medians compared by the Wilcoxon rank-sum test with indicated *p*-value. (D–G) With periodic boundary conditions and uniform excitatory drive, attractor bumps and activity wavefronts appear throughout the neural sheet. (D,E) Excitatory drive during runs and idle periods. (F,G) Snapshots of neural activity during runs and idle periods showing S and inhibitory populations. Each pixel is a neuron and dark colors indicate recent spiking. Red circles indicate regions of recording. Scale bars, 50 neurons. The online version of this article includes the following video and figure supplement(s) for figure 8:

**Figure supplement 1.** Replay properties in simulations with lower velocity gain.
**Figure supplement 2.** Replay properties in simulations with periodic boundary conditions and uniform excitatory drive.
**Figure supplement 3.** Phase precession and theta sequence properties in simulations with periodic boundary conditions and uniform excitatory drive.
**Figure 8—video 1.** Replays preferentially radiate from attractor bumps encoding the current animal position.
https://elifesciences.org/articles/46351#fig8video1
**Figure 8—video 2.** Neural activity during runs, idle periods, and allocentric corrections for a simulation with periodic boundary conditions and uniform excitatory drive.
https://elifesciences.org/articles/46351#fig8video2

(*Figure 8—video 1*). Thus, the decoded position tends to start near the actual position of the animal and travel away from it (*Figure 8B*). Accordingly, the decoded starting position of replays are closer to the actual position than the decoded ending position (*Figure 8C*).

However, replays are not all detected immediately after wavefront nucleation; wavefronts traveling for some time may pass through a region of recorded neurons from any direction. Thus, despite the predominance of replays directed away from the actual animal position, some start at the opposite side of the track and travel toward the actual position (*Figure 8B*). This directional bias has not been directly assessed for grid cells, but it appears in experiments on hippocampal replay (*Diba and Buzsáki, 2007*; *Davidson et al., 2009*).

## Periodic boundary conditions and uniform drive produces replays throughout the neural sheet

Our main model has nonperiodic boundary conditions, which requires excitatory drive to decrease towards the edge of the neural sheet to prevent boundary effects that disrupt path integration (*Burak and Fiete, 2009*). During idle periods, the persistence of attractor bumps at the center of the sheet maintain a representation of the animal's position (*Figure 7B*). With periodic boundaries that eliminate edges, we can implement uniform excitatory drive during both runs and idle periods (*Figure 8D,E*). This allows the entire neural sheet to exhibit attractor bumps during runs and activity wavefronts during idle periods (*Figure 8F,G* and *Figure 8—video 2*). Thus, replays can be recorded from anywhere on the neural sheet. Note that allocentric input at the end of each idle period is essential in this case because positional information in the form of attractor bumps completely disappears. Simulations with periodic boundary conditions and uniform drive produce all the essential replay and theta-related phenomena observed in our main model (*Figure 8—figure supplement 2* and *Figure 8—figure supplement 3*).

# Discussion

## Summary of our results

Decades of hippocampal experiments have established theta phase precession, theta sequences, and replay as central features of place cell dynamics (*O'Keefe, 1976*; *Buzsáki, 1986*; *Skaggs et al., 1996*; *Lee and Wilson, 2002*; *Foster and Wilson, 2006*). Evidence has been accumulating that grid cells, discovered more recently, also exhibit these phenomena (*Hafting et al., 2008*; *Ólafsdóttir et al., 2016*; *O'Neill et al., 2017*). These three phenomena appear to be similar to one another in many ways, but experiments have also demonstrated important discordances. Theta sequences have been understood as the direct consequence of individually phase-precessing neurons (*O'Keefe and Recce, 1993*; *Skaggs et al., 1996*). However, place cell experiments show that phase precession can appear without theta sequences (*Feng et al., 2015*). Theta sequences and replays both represent temporally compressed trajectories, and both seem important for memory consolidation and planning (*Foster and Knierim, 2012*; *Zielinski et al., 2018*). Yet, replay trajectories are significantly faster than theta sequences, and theta sequences only propagate in the direction of motion whereas replays can propagate in any direction relative to the resting animal (*Davidson et al., 2009*; *Feng et al., 2015*; *Zheng et al., 2016*; *O'Neill et al., 2017*).

We unify phase precession, theta sequences, and replay under a single model while maintaining distinctions among them. Phase precession and theta sequences arise naturally from oscillations in attractor bump size and speed, respectively, which are both consequences of medial septum input. However, bump size and speed are not generally correlated across changes in other model parameters (*Figure 6—figure supplement 1*). Thus, it is conceivable that one theta-related phenomenon can exist without the other under certain conditions. Replays in our model arise from the disappearance of attractor bumps and their replacement by traveling wavefronts of activity. Wavefront propagation involves dynamical processes that are fundamentally different from attractor bump motion, and thus has led to different decoded speeds and directions for replays and theta sequences.

Crucially, we produce these phenomena in a continuous attractor network that can still generate 2D grid cells through path integration. Our implementation shares similarities with an early version of the model (*Burak and Fiete, 2009*) and a version whose connectivity can be learned (*Widloski and Fiete, 2014*). However, there are three key differences. First, we use fully spiking neurons, which are rarely used in grid cell attractor models—we have only found them utilized in *English (2017)* and *Bush and Burgess (2014)*, the latter of which combines an attractor network with the oscillatory interference model for grid cells (*Burgess et al., 2007*; *Hasselmo et al., 2007*).

Typically, neural dynamics are founded on firing rates, which can be used to generate spike trains through a memoryless Poisson process (*Burak and Fiete, 2009*; *Widloski and Fiete, 2014*); these models do not capture the strong short-time anticorrelation in spiking due to the reset of membrane potential. Second, we include excitatory-to-excitatory connections in accordance with their recent experimental discovery (*Fuchs et al., 2016*; *Winterer et al., 2017*; *Schmidt et al., 2017*; *Zutshi et al., 2018*). These connections allow wavefronts to propagate among excitatory neurons while nearby inhibitory neurons remain silent. Third, we vary the external input to the network between active and quiescent behavioral states in accordance with observations (*Steriade et al., 2001*; *Simon et al., 2006*). Consequently, the network can instantly switch between operating regimes that exhibit either theta-related phenomena or replays.

Our model presents a distinct conception of entorhinal replay that can be tested experimentally. Direct observation of propagating wavefronts can be pursued by calcium imaging of the MEC (*Heys et al., 2014*; *Diehl et al., 2019*). Otherwise, tetrode recordings of grid cell replay in open field environments may be analyzed for wavefront dynamics. In two dimensions, wavefronts would not represent a single trajectory, but instead a manifold of positions that sweep out a plane through space. New decoding techniques that permit the grid cell population to simultaneously represent inconsistent locations would be required to distinguish between these two possible representations of motion.

## Relationships to experiments and other models

We chose to build a continuous attractor model that can produce theta-related behavior and replay with as simple an architecture as possible, but we imagine that more complex network geometries are possible. For example, during quiescence, our model maintains activity bumps at the center of network and only exhibits replays at the edges; with alternative implementations of excitatory drive, multiple areas may contain persistent attractor bumps that nucleate wavefronts. We have also ignored other known components of the grid system. We do not model multiple grid modules (*Stensola et al., 2012*; *Urdapilleta et al., 2017*; *Kang and Balasubramanian, 2019*). During runs, spike frequency adaptation (*Yoshida et al., 2013*) would reduce the number of spikes late in the grid field, which could enhance phase precession according to the conceptual model presented in *Figure 4* and increase correlation magnitude. During quiescence, the MEC exhibits Up and Down states marked by high and low levels of activity; perhaps transitions between these states, which is associated with CA1 ripples (*Hahn et al., 2012*), can nucleate multiple wavefronts that propagate in synchrony. Moreover, simulation features used to study gamma oscillations in grid cells (*Nolan, 2018*; *Chrobak and Buzsáki, 1998*; *Colgin et al., 2009*) can be introduced to elucidate the role of gamma during theta-related phenomena and replays (*Pfeiffer and Foster, 2015*; *Ramirez-Villegas et al., 2018*).

Our model does not explicitly implement biophysical distinctions between LII and LIII neurons or between stellate and pyramidal cells within LII. These neural subpopulations have different characteristic ranges of phase precession that even vary between experimental reports. *Ebbesen et al. (2016)* found the median precession range of LII stellate and pyramidal cells to be $\approx 130°$ and $\approx 50°$, respectively, which is smaller than the values of $\approx 110°$ and $\approx 170°$ found by *Reifenstein et al. (2016)*. Meanwhile, LIII grid cells, which often possess conjunctive head direction tuning (*Sargolini et al., 2006*) like the neurons in our model, predominantly exhibit phase-locking with phase precession range close to 0° according to *Hafting et al. (2008)*. However, *Reifenstein et al. (2014)* reports small, but significant precession in LIII, and single-lap analyses by *Ebbesen et al. (2016)* finds that LIII neurons precess even more than LII pyramidal cells. *O'Neill et al. (2017)* does not distinguish between these cell types and observes all of these phase behaviors; this heterogeneity is dynamically captured by the excitatory principal cells in our model. Simulations implementing multiple types of principal cells would be required to precisely assign the distinctions in theta phase properties to the appropriate cell type.

Other models have been proposed for theta phase precession (*Tsodyks et al., 1996*; *Thurley et al., 2008*; *Navratilova et al., 2012*; *Thurley et al., 2013*) and replay (*Ponulak and Hopfield, 2013*; *Azizi et al., 2013*; *Bayati et al., 2015*; *Chenkov et al., 2017*; *Gönner et al., 2017*; *Haga and Fukai, 2018*). For example, one grid cell model uses after-spike depolarization within a 1D continuous attractor network to generate phase precession and theta sequences (*Navratilova et al., 2012*). This experimentally observed feature pulls attractor bumps back to

recently active locations every theta cycle. A place cell model combines an oscillating membrane potential, as found in our model, with synaptic facilitation (*Thurley et al., 2008*). These features could be added to our model to strengthen theta-related phenomena and potentially increase correlation magnitude, but they not required. Previously proposed replay models were all intended to address place cells, not grid cells. Several of them encode replay trajectories into synaptic weights, either through hard-wiring or a learning mechanism (*Chenkov et al., 2017*; *Gönner et al., 2017*; *Haga and Fukai, 2018*). Two models have suggested that replays originate from wavefronts of activity propagating through networks of place cells (*Ponulak and Hopfield, 2013*; *Bayati et al., 2015*; *Palmer et al., 2017*). These wavefronts then enhance connections through the network though spike-timing-dependent plasticity rules, which could account for the ability of reward to modulate hippocampal replay (*Pfeiffer and Foster, 2013*; *Ambrose et al., 2016*; *Palmer and Gong, 2013*).

## Possible implications

Although phase precession, theta sequences, and replay arise without learning in our model, they are not necessarily epiphenomenal in the sense that they serve no cognitive purpose. Grid cells are thought to provide a spatial representation that is immediately available upon entering an environment and that remains stable over days of experience (*Hafting et al., 2005*; *Leutgeb et al., 2007*; *Diehl et al., 2019*). They provide spatial information to the hippocampus (*Cheng and Frank, 2011*; *Hales et al., 2014*), whose place cells associate location with environmental context (*Wilson and McNaughton, 1993*; *Anderson and Jeffery, 2003*) and reward (*Hollup et al., 2001*; *Mamad et al., 2017*) to aid the animal in pursuing its goals. In this vein, we envision that superficial MEC layers act as a pattern generator for phase precession, theta sequences, and replay, especially in new environments. These phenomena are then presented to the hippocampus to be refined by experience and modulated by reward.

The concept that sequences in hippocampus arise from superficial MEC layers has been experimentally substantiated for theta phenomena: lesioning the MEC disrupts phase precession and theta sequences in place cells (*Schlesiger et al., 2015*), but grid cell phase precession is not affected by hippocampal inactivation (*Hafting et al., 2008*). Driving place cell replays with grid cell replays would require temporal coordination between the two, for which experimental support is mixed. *O'Neill et al. (2017)* does not find coordination between grid cell replays in superficial MEC layers and sharp-wave-ripples (SWRs) in CA1, during which place cell replays appear. However, *Ólafsdóttir et al. (2016)* and *Ólafsdóttir et al. (2017)* report that hippocampal replays represent trajectories coherently with grid cell activity in deep MEC layers, and *Yamamoto and Tonegawa (2017)* reports that SWRs in CA1 are preceded by SWRs in superficial MEC layers. Further investigation would help to elucidate the temporal relationships between hippocampus and MEC. Also, note that we do not exclude the possibility that place cell replays can also be initiated within the hippocampus or by subcortical nuclei (*Stark et al., 2014*; *Schlingloff et al., 2014*; *Oliva et al., 2016*; *Angulo-Garcia et al., 2018*; *Sasaki et al., 2018*).

Place cell activity appears to contain an abundance of certain short sequences that represent contiguous locations in novel environments and participate frequently in replays (*Liu et al., 2018*). This finding is related to the disputed (*Silva et al., 2015*) observation of hippocampal preplay (*Dragoi and Tonegawa, 2011*; *Grosmark and Buzsáki, 2016*; *Liu et al., 2019*). From the perspective of our model, such preexisting sequences may arise from wavefronts generated intrinsically by grid cells, which then drive corresponding place cells. After experience, hippocampal plasticity supplements the sequences with additional place cells to enhance their representation of meaningful trajectories (*Grosmark and Buzsáki, 2016*; *Drieu et al., 2018*). In this way, wavefronts in the MEC can accelerate learning by providing a scaffold for the hippocampus to modify and improve. Meanwhile, the MEC maintains a symmetric representation of space, so the next experience, possibly with different spatial properties, can be learned equally well.

Yet, we emphasize that our model focuses on the mechanistic origins of grid cell sequences and its results hold regardless of whether the suggested relationships between MEC and hippocampus are borne out by future experiments.

## Materials and methods

This section provides an overview of our simulation methods. Full details are provided in Appendix 1.

### Model architecture and dynamics

Our architecture is inspired by *Burak and Fiete (2009)* and *Widloski and Fiete (2014)*. A 2D neural sheet contains five overlapping populations, each with neurons at $\mathbf{r} = (x, y)$, where $x = 1, \ldots, n$ and $y = 1, \ldots, n$. There are four excitatory populations: N, S, W, and E; we index them by $P$ and use the symbol $+$ for parameters common to all of them. There is one inhibitory population $-$.

Each excitatory neuron is described by a membrane potential $\phi^P(\mathbf{r}, t)$ and a spike indicator $s^P(\mathbf{r}, t)$, which equals either 1 if neuron $\mathbf{r}$ in population $P$ spiked at time $t$ or 0 otherwise. When the potential exceeds a threshold of 1 in arbitrary units, the neuron fires a spike and its potential is reset to 0:

$$\phi^P(\mathbf{r}, t) \leftarrow 0 \quad and \quad s^P(\mathbf{r}, t) \leftarrow 1. \tag{1}$$

We prevent $\phi^P(\mathbf{r}, t)$ from decreasing past –1 as a limit on hyperpolarization. The same definitions and dynamics apply to the inhibitory population $-$.

Membrane potentials follow leaky integrator dynamics. The excitatory neurons obey

$$\tau_m^+ \frac{\phi^P(\mathbf{r}, t+\Delta t) - \phi^P(\mathbf{r}, t)}{\Delta t} + \phi^P(\mathbf{r}, t)$$
$$= \sum_{P', \mathbf{r}'} w^+(\mathbf{r} - \mathbf{r}' - \xi\hat{\mathbf{e}}^{P'}) s^{P'}(\mathbf{r}', t - \tau_s^{++}) + \sum_{\mathbf{r}'} w^-(\mathbf{r} - \mathbf{r}') s^-(\mathbf{r}', t - \tau_s^-)$$
$$+ a^+(\mathbf{r})\left[1 + \alpha\hat{\mathbf{E}}^P \cdot \mathbf{V}(t)\right] + \zeta^P(\mathbf{r}, t). \tag{2}$$

And the inhibitory neurons obey

$$\tau_m^- \frac{\phi^-(\mathbf{r}, t+\Delta t) - \phi^-(\mathbf{r}, t)}{\Delta t} + \phi^-(\mathbf{r}, t)$$
$$= \sum_{P', \mathbf{r}'} w^+(\mathbf{r} - \mathbf{r}' - \xi\hat{\mathbf{e}}^{P'}) s^{P'}(\mathbf{r}', t - \tau_s^{-+}) + a^-(t) + \zeta^-(\mathbf{r}, t). \tag{3}$$

Neural drive is given by $a$, which is modulated by the animal's velocity $\mathbf{V}(t)$ for excitatory neurons. Each excitatory population $P$ prefers a perpendicular direction in space $\hat{\mathbf{E}}^P$, which we call North, South, West, and East for the populations N, S, W, and E. Neural connectivity is given by $w$. Each excitatory population $P$ has its synaptic outputs shifted by a small amount $\xi$ in a preferred direction $\hat{\mathbf{e}}^P$ on the neural sheet, which we call up, down, left, and right for the populations N, S, W, and E. Spikes produce an instantaneous change in postsynaptic membrane potential after a synaptic delay $\tau_s$, which is longer for excitatory-to-excitatory connections ($\tau_s^{++}$) than for excitatory-to-inhibitory connections ($\tau_s^{-+}$) based on axon morphology (*Schmidt et al., 2017*). Independent, zero-mean, normally distributed noise $\zeta$ is added to each neuron at each timestep. See *Table 1* for complete variable definitions and values.

### Drive and connectivity

The drive to excitatory populations is

$$a^+(\mathbf{r}) = \begin{cases} a_{min}^+ + (a_{max}^+ - a_{min}^+)\frac{1+\cos(\pi\rho/\rho_{a+})}{2} & \rho < \rho a+ \\ a_{min}^+ & \rho \geq \rho a+, \end{cases} \tag{4}$$

where $\rho = \sqrt{\left(x - \frac{n+1}{2}\right)^2 + \left(y - \frac{n+1}{2}\right)^2} / \frac{n}{2}$ is a scaled radial distance for the neuron at $\mathbf{r} = (x, y)$. It equals 0 at the center of the neural sheet and approximately one at the midpoint of an edge. The drive to inhibitory subpopulations is

$$a^-(t) = a_{mag}^- - a_{th}^- \cos(2\pi ft + \psi_0), \tag{5}$$

where $\psi_0$ is a phase offset chosen randomly at the start of each lap.

The synaptic connectivity from excitatory neurons to all neurons is based on the symmetric function

$$w^+(\mathbf{r}) = \begin{cases} w_{mag}^+ \frac{1+\cos(\pi r/r_{w^+})}{2} & r < rw+ \\ 0 & r \geq rw+, \end{cases} \tag{6}$$

where $r = |\mathbf{r}|$. The synaptic connectivity from inhibitory neurons to excitatory neurons is

$$w^-(\mathbf{r}) = \begin{cases} -w_{mag}^- \frac{1-\cos(\pi r/r_{w^-})}{2} & r < 2rw- \\ 0 & r \geq 2rw-. \end{cases} \tag{7}$$

There is no connectivity from inhibitory neurons to other inhibitory neurons.
See *Table 1* for complete variable definitions and values.

## Brief allocentric correction

The grid-like pattern of excitatory drive during brief allocentric corrections (*Figure 2B*) represents inputs from neurons with allocentric responses, such as border, boundary vector, or landmark cells (*Evans et al., 2016*; *Raudies and Hasselmo, 2015*; *Savelli et al., 2017*). Interactions between allocentric neurons and grid cells have also been implemented in other continuous attractor models (*Ocko et al., 2018*; *Keinath et al., 2018*). In our model, allocentric input is learned in an accelerated manner during simulation setup. Immediately before the main simulation phase, we simulate one run in each track direction followed by a learning period at either end of the track. The locations of attractor bumps during learning periods are stored by tallying the number of spikes produced at each neural sheet location. These tallies are linearly rescaled to serve as the allocentric drive to excitatory populations.

## Single neuron recording and decoding

We simulate our model 10 times with slightly different run trajectories randomly generated by fractional Brownian motion processes, followed by 10 more times with the same trajectories but reversed velocities. These simulations produce spike trains for all neurons in the network. To match multiunit tetrode recordings, we select two sets of up to 150 excitatory neurons in each simulation. Each of these sets is an independent recording to be analyzed separately, which we call a 'simulation' in our results.

We select neurons for recording as follows. For the first set, we choose four points on the neural sheet whose distance from the center is 95 neurons and whose angular positions, or clock positions, are random but equally spaced. From circular areas of radius 12 centered at these points, we randomly select up to 150 excitatory neurons whose firing fields are stable across laps. For the second set, we choose the four circular areas with the same distance from the center of the sheet and with angular positions offset 45° from the first four areas. We again select up to 150 neurons with stable firing fields. All single-simulation results reported in the main text come from the same simulation.

We use grid cell firing fields accrued from all laps in both directions to decode the animal's position from the population activity of its recorded neurons. We use the standard Bayesian decoding procedure that assumes independent, Poisson-distributed spike counts and a uniform prior (*Zhang et al., 1998*; *O'Neill et al., 2017*).

## Acknowledgements

The authors thank John Widloski, David Foster, Fritz Sommer, Zengyi Li, Guy Isely, Kate Jeffery, Hugo Spiers, and Srdjan Ostojic for stimulating discussions and helpful ideas. Part of this work was performed at the Simons Institute for the Theory of Computing.

## Additional information

### Funding

| Funder | Grant reference number | Author |
|---|---|---|
| Adolph C. and Mary Sprague Miller Institute for Basic Research in Science, University of California Berkeley | Postdoctoral fellowship | Louis Kang |
| Army Research Office | W911NF-13-1-0390 | Michael R DeWeese |

The funders had no role in study design, data collection and interpretation, or the decision to submit the work for publication.

### Author contributions

Louis Kang, Conceptualization, Software, Investigation, Visualization, Methodology, Writing—original draft, Writing—review and editing; Michael R DeWeese, Resources, Supervision, Funding acquisition, Visualization, Writing—original draft, Writing—review and editing

### Author ORCIDs

Louis Kang (iD) https://orcid.org/0000-0002-5702-2740

### Decision letter and Author response

Decision letter https://doi.org/10.7554/eLife.46351.sa1
Author response https://doi.org/10.7554/eLife.46351.sa2

## Additional files

### Supplementary files

• Source code 1. Source code for the main simulation written in C.

• Transparent reporting form

### Data availability

Source code for the simulations have been included as supporting files.

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

## Appendix 1

### Additional methods

#### Model details

##### Setup

We initialize each neuron on the neural sheet to a random membrane potential between 0 and 1. We evolve the network according to *Equation 2* and *Equation 3* for 500 timesteps without velocity input to generate a rough grid-like pattern of activity. To make the grid more regular and to anneal grid defects, we perform three evolutions of timesteps with constant velocity input **V** of magnitude 0.5 m/s and angles, $\pi/2 - \pi/5$, $2\pi/5$, and $\pi/4$ successively. We then evolve the network for 4 laps without idle periods.

Finally, we learn allocentric inputs as follows. We simulate 2 additional laps with 1000 timesteps of idle periods at each end of this track. During this time, both the excitatory and inhibitory drives are maintained at their run values without theta oscillations. Over the final 500 timesteps at each end, the number of spikes produced at each neural sheet location across the 4 excitatory populations are tallied. These values are linearly rescaled to serve as allocentric inputs to excitatory neurons, with the highest spike count corresponding to $a_{\mathrm{max}}^+$ and anything below the 5th percentile of nonzero activity corresponding to $a_{\mathrm{min}}^+$. Then the main simulation phase begins.

##### Simulations with periodic boundary conditions

Since annealing grid defects is difficult with periodic boundary conditions, we initialize membrane potentials with a grid-like configuration on the neural sheet as in *Burak and Fiete (2009)*. The grid scale is chosen to be similar to that self-organized without periodic boundary conditions. The rest of simulation setup proceeds similarly as above, except the three evolutions with constant velocity input last only 500 timesteps each.

Parameter values that differ from those in *Table 1* are $n$ = 192, $a_{\mathrm{mag}}^+ = 1.2$ during runs and 0.92 when idle for the uniform excitatory drive, $a_{\mathrm{max}}^+ = 1.4$ during allocentric corrections, $a_{\mathrm{mag}}^- = 0.75$ during runs and –0.1 when idle, and $a_{\mathrm{th}}^- = 0.15$ during runs. Results are shown in *Figure 8F, G*.

##### Trajectory

For 2D open field simulations, we use velocity inputs extracted by *Burak and Fiete (2009)* from a real animal trajectory reported in *Hafting et al. (2005)*.

For 1D track simulations, we generate trajectories as follows. Each trajectory consists of 36 runs alternating in direction, each lasting 1.5 s. Each run is followed by an idle period of 1.5 s and an allocentric correction period of 0.5 s. The velocity angle is $\pi/5$, chosen to avoid directions of symmetry in the neural sheet.

In each run, the speed linearly increases from 0 m/s to 0.5 m/s over the first 0.3 s, remains at 0.5 m/s over the next 0.9 s, and linearly decreases from 0.5 m/s to 0 m/s over the last 0.3 s. To the middle 0.9 s of constant speed, we add noise. The noise is generated by a fractional Brownian motion process with drift 0, volatility 1, and Hurst index 0.8. We then add an overall constant at each timestep to make the noise sum to 0 (and thus, to leading order, contribute 0 additional displacement) and introduce an overall scale factor to make its maximum magnitude 0.1 m/s. We add this processed noise to the constant velocity of 0.5 m/s. The initial theta phase of each run is chosen randomly, and the phase increases during runs at a constant rate to span 360˚ over 125 ms.

Excitatory and inhibitory drives change between runs and idle periods according to *Table 1*. They do so by linearly interpolating between corresponding values over the first 0.3 s of each idle period.

## Single neuron recording and position decoding

### Selection of neurons for recording

As described in Materials and methods, we select up to 150 neurons total for recording from four equally spaced circular areas on the neural sheet (up to 200 neurons total for simulations with periodic boundary conditions). For the main simulation, these areas are centered 95 neurons from the center of the sheet; for the simulations with lower velocity gain in *Figure 8A-C*, the distance is 75 neurons; for the simulations with periodic boundary conditions in *Figure 8D-G*, the distance is 60 neurons. In each area, we randomly choose 30–50 neurons from each excitatory population and compute their firing fields for each run as follows. We partition space into 5 cm bins and compute the firing rate of each bin as the number of spikes divided by the occupancy time; these raw fields are smoothed by a Gaussian filter of width 4 bins. Firing fields for leftward and rightward runs are considered separately. In at least one direction, recorded neurons must have maximum firing rate above 0.5 Hz for all runs, and the normalized correlation, defined as $\mathbf{u} \cdot \mathbf{v}/|\mathbf{u}||\mathbf{v}|$ for two vectors $\mathbf{u}$ and $\mathbf{v}$, must be above 0.6 for all pairs of runs.

### Firing fields

For 2D open field simulations, we partition space into bins indexed by $\mathbf{R} = (X, Y)$ and compute the firing rate of each bin as the number of spikes divided by the occupancy time. These raw fields are smoothed by a Gaussian filter of width 5 bins to produce the field map $F(\mathbf{R})$. We define the normalized autocorrelation map as

$$C(\mathbf{R}) = \frac{\frac{1}{N(\mathbf{R})} \sum_{\mathbf{R}'} F(\mathbf{R}')F(\mathbf{R}' + \mathbf{R})}{\frac{1}{N(0)} \sum_{\mathbf{R}'} F(\mathbf{R}')F(\mathbf{R}')}, \tag{8}$$

where $N(\mathbf{R})$ is the number of pairs of positions separated by $\mathbf{R}$. Autocorrelation maps are shown in *Figure 1E*.

For 1D track simulations, we partition space into 2 cm bins indexed by $X$ and compute the firing rate of each bin as the number of spikes divided by the occupancy time over all laps. These raw fields are smoothed by a Gaussian filter of width 1 bin to produce the field map $F(X)$. Field maps are shown in *Figure 2D*.

To categorize a simulation as exhibiting 'one field' or 'two fields,' we first order all neurons by the track position at which they are most active. We exclude the middle 40% of all neurons, since we do not expect neurons with central field positions to exhibit two fields. We then shift each field such that its position of peak activity is at 0 cm and calculate the fraction of field activity located more than 18 cm away. If this fraction is greater than 0.1, we consider the simulation as exhibiting two fields. The center of mass of this distant activity, using the absolute value of position, is the distance between fields for the simulation.

### Position decoding

We use the standard Bayesian techniques to decode position from the population activity of grid cells (*Zhang et al., 1998*; *O'Neill et al., 2017*). Let $s_k(t)$ count the number of spikes fired by the recorded neuron $k$ within a time window of $\Delta T$ around $t$. Each neuron has a field map $F_k(X)$ as described above. For every time $t$, we wish to calculate the probability that the animal is at position $X$ given the population activity $\mathbf{s}(t)$. We do so through Bayes's rule, assuming a flat prior and a spiking likelihood that obeys independent Poisson statistics:

$$p(X|\mathbf{s}(t)) \propto p(\mathbf{s}(t)|X) = \prod_k p(s_k(t)|X) \propto \left( \prod_k F_k(X)^{s_k(t)} \right) \exp\left( -\Delta T \sum_k F_k(X) \right). \tag{9}$$

We increment time in steps of 5 ms. For theta-related analysis, we use a sliding decoding window of width $\Delta T = 20$ ms. For analysis of replays, whose timescales are shorter and spikes are more concentrated, we use $\Delta T = 10$ ms and require each window to contain at least two spikes.

## Theta-related analysis

### Multiunit activity

To calculate the multiunit activity averaged over theta cycles, we sum spike trains over recorded neurons during each run, extract the multiunit activity of each complete theta cycle, and average these extracts. This theta-modulated multiunit activity is shown in **Figure 2E**.

### Phase precession

We use circular-linear regression and correlation as described in **Kempter et al. (2012)** to quantify the relationship between theta phase and position within a grid field. For each neuron, we combine spikes across all rightward runs and across all leftward runs, considering each direction separately. For each direction, we identify a separation between grid fields as a 10 cm span without spikes. For each neuron and each direction, we only consider the most central field. We exclude fields that spike within 3 cm from the track ends to avoid edge effects, and also exclude fields containing fewer than 30 spikes or spanning less than 12 cm wide.

For each field, we record the position $X_j$ and theta phase $\psi_j$ at which each of its spikes $j$ occurred. We hypothesize that the phase of each spike is linearly influenced by its position:

$$\psi_j = 2\pi q X_j + \psi_0, \tag{10}$$

where $q$ is the phase precession slope and $\psi_0$ is the phase offset. We wish to fit these parameters by minimizing the total circular distance between true phases and corresponding predicted phases. The circular distance between two phases $\psi$ and $\theta$ is defined to be $2[1 - \cos(\psi - \theta)]$. As explained in **Kempter et al. (2012)**, we can transform the minimization over into the following maximization:

$$\hat{q} = *arg\,max_q \sqrt{\left[\frac{1}{S}\sum_j \cos(\psi_j - 2\pi q X_j)\right]^2 + \left[\frac{1}{S}\sum_j \sin(\psi_j - 2\pi q X_j)\right]^2}, \tag{11}$$

where is the number of spikes. We perform the maximization numerically over $q$. The maximum value is called the mean resultant length (or vector strength), and it is used as our fit score. The minimizing value of $\psi_0$ can be then calculated directly:

$$\hat{\psi}_0 = \arctan\frac{\sum_j \sin(\psi_j - 2\pi\hat{q}X_j)}{\sum_j \cos(\psi_j - 2\pi\hat{q}X_j)}. \tag{12}$$

Regression results are shown in **Figure 3D, E**.

We can also estimate the circular-linear correlation coefficient, which is analogous to the Pearson correlation coefficient for two linear variables. Note that since the marginal distribution of $\Theta$ may be uniform, we use the following expression (**Kempter et al., 2012**):

$$\hat{\rho} = \frac{\left|\sum_j e^{i(\psi_j - \Theta_j)}\right| - \left|\sum_j e^{i(\psi_j + \Theta_j)}\right|}{2\sqrt{\sum_j \sin^2(\psi_j - \bar{\psi})}\sqrt{\sum_j \sin^2(\Theta_j - \bar{\Theta})}}, \tag{13}$$

where $\Theta_j = 2\pi|\hat{q}|X_j$ is a circular variable related to $X_j$, and $\bar{\psi} = \arctan(\sum_j \sin\psi_j / \sum_j \cos\psi_j)$ and $\bar{\Theta} = \arctan(\sum_j \sin\Theta_j / \sum_j \cos\Theta_j)$ are respective circular means. This correlation coefficient is shown in **Figure 3C**.

### Normalized bump coordinate

Over the course of a simulation, attractor bumps trace out trails of activity on the neural sheet (**Figure 2C**). For a neuron at a given sheet location, we quantify how centrally an attractor bump passes through it with a metric called the normalized bump coordinate. We first tally the total number of spikes occurring at each location on the neural sheet across all excitatory

populations. This yields an overall firing rate for each sheet location. For a neuron at a given location, we look for the central axis of the nearest attractor bump trail as follows. We find the highest overall firing rate in a window of length 10 neurons and width 4 neurons centered at the given location and oriented with long side perpendicular to the track orientation. The normalized bump coordinate is the ratio between the given location's overall firing rate and this highest overall firing rate, which should correspond to an attractor bump's central axis. Bump coordinate results are shown in *Figure 3—figure supplement 4G*.

## Simplified phase precession models

For our conceptual model with bump size oscillations, we imagine an attractor bump moving through a recorded neuron. Although the bump naturally oscillates in speed, we advance the bump with constant speed in the neural sheet; similar results (not shown) are obtained if the speed oscillations depicted in *Figure 6B* are included. We stipulate that the middle of the bump passes through the recorded neuron at position 0 and time 0. Thus, the 1D profile of the bump is given by the region between

$$b_{\pm}(t) = \bar{v}t \pm [\bar{b} + \Delta b \cos(2\pi f t - \psi_0)], \tag{14}$$

where $\bar{b} - \Delta b$ and $\bar{b} + \Delta b$ are the smallest and largest radii of the bump, $\bar{v}$ is the constant bump velocity, $f$ = 8 Hz, and the phase offset $\psi_0$ is incremented by $\pi/5$ for every new lap. We choose $\bar{v} = 17\,\text{neurons/s}$, $\bar{b} = 3.3\,\text{neurons}$, and $\Delta b$ = 1.4 neurons, which are close to the values presented in *Figure 4B* and *Figure 6B*. Whenever $b_-(t) \leq 0 \leq b_+(t)$, the neuron fires a spike, subject to a refractory period of 40 ms. Results are shown in *Figure 4C*.

For our simplified model with average bump dynamics, we first determine the average bump shape as a function of theta phase as described later. As in the conceptual model described immediately above, we imagine the central axis of this average bump moving through a recorded neuron. We simulate 18 laps with evenly spaced initial theta phases. The recorded neuron's spiking rate is proportional to bump activity. We can linearly rescale bump activity to a desired maximum level across all theta phases; that is, the bump would still exhibit different levels of maximum activity at each theta phase. Spikes are generated with sub-Poisson statistics using the decimation procedure described in *Burak and Fiete (2009)*; we choose the coefficient of variation to be $1/\sqrt{8}$. We also use a refractory period of 40 ms. Results are shown in *Figure 4E, F*.

## Theta sequences

We visualize theta sequences by averaging the decoded position over theta cycles. From all runs along the track, we take overlapping quadruplets of consecutive theta cycles, discarding those during which the animal comes within of the track ends to avoid edge effects. We assign time 0 to the middle of each theta quadruplet. We then shift the decoded positions of each quadruplet by the actual position of the animal at time 0, such that the origin corresponds to the actual animal position at the middle of the theta quadruplet. Before averaging, we proceed as in *O'Neill et al. (2017)* and divide the probability at each timestep by its maximum value. We then separately average each timestep of the quadruplet, ignoring cycles without spikes within that timestep.

To find the slope of the fastest decoded motion within the theta cycle, we use a similar approach to *O'Neill et al. (2017)*. For each time window whose midpoint lies within the third cycle of the averaged theta quadruplet, we find the fit line that captures the most decoded probability. In *O'Neill et al. (2017)*, this involves summing all the probability that lies within a certain number of position bins away from each possible line. Instead of setting such a hard cutoff, we pass the averaged probability distribution at each timestep through a Gaussian filter of width. For each candidate line, we sum over the filtered probability of the closest position bin at each timestep; this process is analogous to weighting each bin according to a Gaussian function of its distance to the line and summing the unfiltered probability over all bins. The fit line is the one with the largest total probability. Finally, across all time windows, we choose the fit line with highest slope to represent the averaged theta sequence. These sequences and the Gaussian-filtered, averaged probability distribution is shown in *Figure 5B*.

## Replay analysis

### Replay detection

As in *O'Neill et al. (2017)*, we look for replays within highly synchronous events (HSEs). To detect HSEs, we first filter the multiunit activity during idle periods by a Gaussian of width and filter size, and we remove all zero-valued timesteps. HSEs are time intervals lasting at least during which the processed multiunit activity exceeds its 20th percentile and contains at least one timestep that exceeds its 80th percentile (or 60th percentile for simulations with periodic boundary conditions).

For each HSE, we look for linear replays by finding the fit line that captures the most decoded probability as we did for theta sequences. We first pass the probability distribution at each timestep during the HSE through a Gaussian filter whose width is and whose maximum value is 1. For each candidate line, we sum over the filtered probability of the closest position bin at each timestep and divide by the number of timesteps that the line spends within track limits. The result is an average probability captured by the line, which we call a replay score. For each HSE, the fit line is the one with the highest replay score. Finally, we exclude HSEs whose fit lines have a replay score less than 0.6 or spend less than 30 ms or 30 cm within track limits. The remaining fit lines are replays. Extracted replays are shown in *Figure 7E*.

For simulations whose neurons exhibit two fields, the probability distribution of the decoded position may be spread over both fields. Thus, we supplement each candidate line with two more lines of the same slope. These three lines are separated by the distance between grid fields. We sum over the filtered probability of the closest position bins to each of these lines.

### Replay properties

The speed of a replay is the magnitude of the slope of its fit line, as reported in *Figure 7G*.

The start position of a replay is the fit line position at the beginning of its HSE, which is clipped to 0 cm or 60 cm if it lies outside track limits. Similarly, the end position of a replay is the fit line position at the end of its HSE, which is similarly clipped. These positions are reported in *Figure 8C*.

## Simulations with fixed inhibitory drive and constant velocity

### Setup and trajectory

These simulations are structured similarly to the main simulations on a 1D track. However, we choose a constant inhibitory drive $a_{\mathrm{mag}}^{-}$ and eliminate theta oscillations with $a_{\mathrm{th}}^{-} = 0$. During setup, instead of evolving the network with an animal trajectory, we evolve 1000 timesteps with constant velocity of magnitude 0.5 m/s and angle $\pi/5$. For the main simulation, we evolve another 1000 timesteps with the same velocity.

### Attractor bump speed

To extract the speed of attractor bumps, we first divide time into 40 ms bins. Within each bin, we sum the spikes at each position on the neural sheet across all excitatory populations, and we pass the result through a Gaussian filter of width. We thus obtain smoothed activity maps at 40 ms intervals.

We find the seven activity peaks that remain closest to the center of the neural sheet throughout the simulation. For each time bin, we take the average position of all seven peaks, and fit this average motion with a linear model. The norm of the resulting velocity is the bump speed, as reported in *Figure 6B*.

### Attractor bump size

To extract the size of attractor bumps, we return to the raw spike trains, which we register across the course of the simulation by compensating for the overall bump motion. That is, to each spike emitted by an excitatory neuron, we assign a shifted neural sheet position equal to

the true position of the neuron minus the bump velocity obtained above multiplied by its spiking time.

Around each of the seven activity peaks identified above, which are stationary after registration, we draw a circle of radius 12 neurons and calculate the center of mass of all enclosed spikes over the entire simulation. This point is the center of the registered activity bump. We then calculate the registered distance of each spike to the center; the 90th percentile is taken to be the radius of the activity bump. Bump diameters averaged over the seven activity peaks are reported in *Figure 4B*.

## Attractor bump shape

To determine the average attractor bump shape during runs, we use periodic simulations with uniform excitatory drive to generate a set of attractor bumps that experience the same drive. We reintroduce oscillations into the inhibitory drive. These simulations use $a_{\mathrm{mag}}^+ = 1.2$, $a_{\mathrm{mag}}^- = 0.7$, and $a_{\mathrm{th}}^- = 0.15$.

We first perform registration, as described above, for 12 attractor bumps. We count the number of spikes across all bumps at each registered location on the neural sheet (using 0.2 neuron bins) as a function of time within the theta cycle (using 1 ms bins). We then apply Gaussian filters of width 3 time bins and 5 position bins to obtain average bump shape as a function of theta phase, as shown in *Figure 4D* and *Figure 4—video 2*.

## Appendix 2

# Simulations without brief allocentric correction

### Model details

Simulation setup proceeds similarly to simulations with allocentric correction, except we evolve the network for 6 laps including idle periods after the three evolutions with constant velocity input. We also do not learn allocentric connections.

Without allocentric correction, the main simulation can only be run for 12 laps instead of 36. Each lap consists of a 1.5 s run and a 2.5 s idle period. Excitatory and inhibitory drives linearly interpolate between run and idle values over the first and last 0.3 s of each idle period.

We perform 20 replicate simulations in each direction instead of 10. Parameter values that differ from those in *Table 1* are $a_{\mathrm{mag}}^{-} = 0.65$ during runs and 0.1 when idle, $\xi = 2$, $\alpha = 0.12\,\mathrm{s/m}$, and $\mathrm{Var}[\zeta^{P}(\mathbf{r}, t)] = \mathrm{Var}[\zeta^{-}(\mathbf{r}, t)] = 0.012^{2}$.

### Analysis details

With fewer laps compared to simulations with allocentric correction, we use more lenient cutoffs for both theta-related and replay analysis. During phase relationship analysis, we require a minimum of 15 spikes, instead of 30. We also use a regression fit score cutoff of 0.3, instead of 0.4, below which neurons are considered phase-independent. During replay detection, replays must span a minimum of 20 cm, instead of 30 cm.

### Results

In simulations without brief allocentric corrections, many theta-related and replay properties are present a weaker level. We can only run the simulations for 12 laps, and we must use lower velocity gain so that the attractor bumps do not travel far on the neural sheet (*Appendix 2—figure 1A* and *Appendix 2—video 1*). Otherwise, attractor drift will cause the firing fields to change significantly between the start and the end of the simulation.

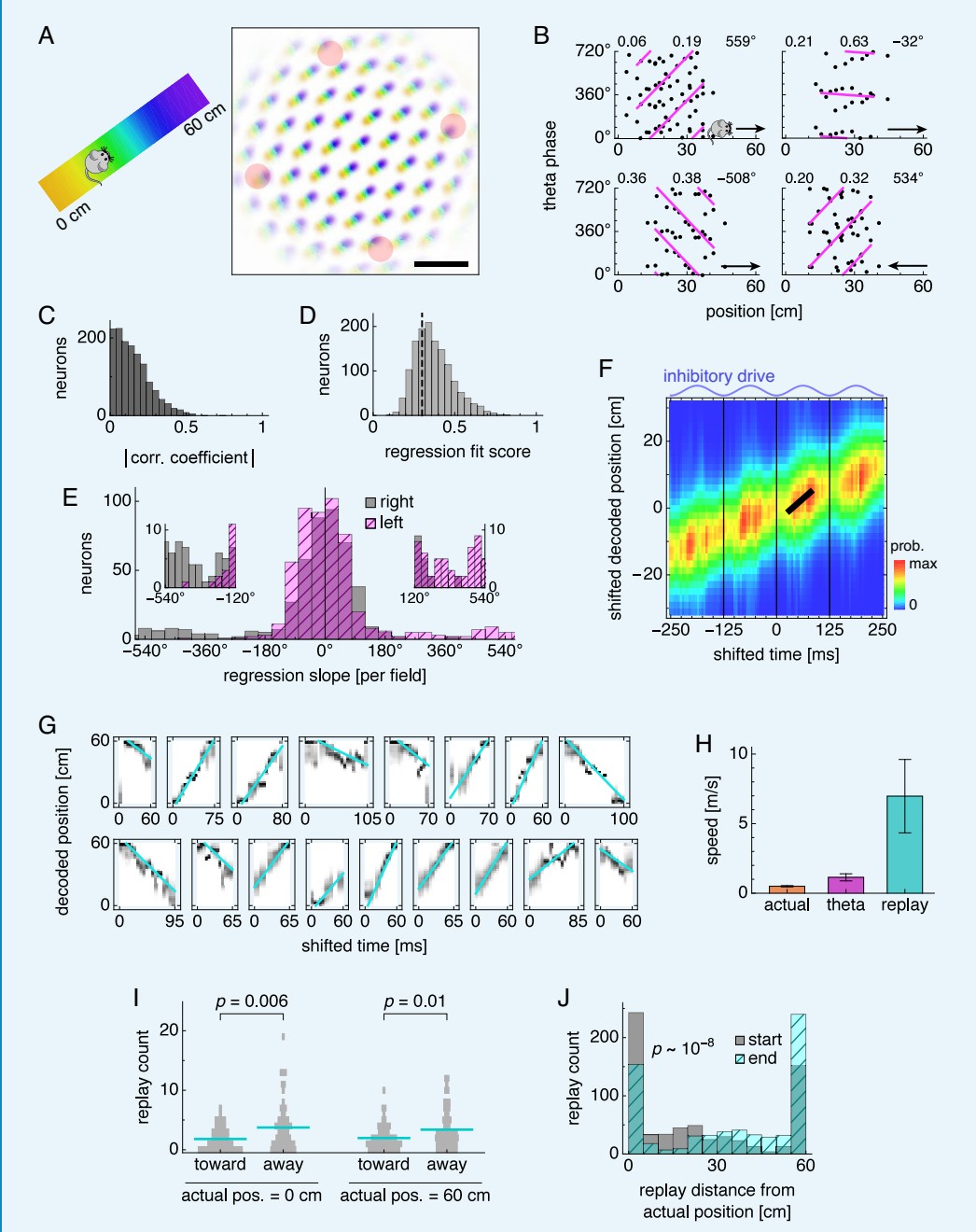

**Appendix 2—figure 1.** Theta-related and replay properties in simulations without brief allocentric corrections. (**A**) Left, track diagram. Right, neural activity over runs with each neuron colored according to the track position at which it attained maximum firing rate. Red circles indicate regions of recording. (**B**) Relationship between animal position and theta phase for representative neurons. Clockwise from top left, relationships depicted are phase independence, phase locking, phase precession, and phase precession. Dots represent spikes during runs in the directions indicated by arrows. Lines indicate fit by circular-linear regression. Numbers in each panel from top left to top right indicate magnitudes of correlation coefficients, regression fit scores, and precession ranges. (**C–E**) Data across all replicate simulations. (**C**) Magnitudes of circular-linear correlation coefficients. Mean ± s.d.: 0.15 ± 0.11. (**D**) Fit scores for circular-linear regression. (**E**) Regression slopes for neurons with fit score >0.3. (**F**) Decoded position averaged over theta cycles. Thick black line fit to theta sequence. (**G**) Replays with cyan fit lines. (**H**) Mean actual run speed (error bars indicate s.d. over time),

mean theta sequence speed (error bars indicate s.d. over replicate simulations), and mean replay speed (error bars indicate s.d. over replays). (**I**) Direction of replay propagation across simulations. Means indicated with lines. *p*-values from the Mann-Whitney *U* test show that a significantly greater number of replays propagate away from either actual position. (**J**) Distances between actual position and decoded position at the start or end of replays. Paired medians compared by the Wilcoxon rank-sum test with indicated *p*-value. Scale bars, 50 neurons.

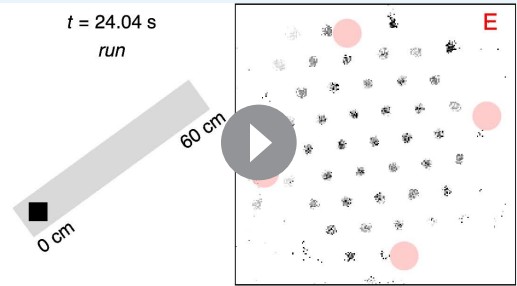

**Appendix 2—video 1.** Neural activity during runs and idle periods during a simulation without allocentric corrections.      Left, position of the animal (black square) along a 1D track. Right, neural activity of the E population. Each pixel is a neuron, with black corresponding to current spikes and lightest gray corresponding to spikes 40 ms ago. Red circles indicate regions of recording.

During runs, our neurons can still be divided into subgroups whose activity exhibits different phase relationships (*Appendix 2—figure 1B*). The magnitude of circular-linear correlation (*Appendix 2—figure 1C*; 0.15 ± 0.11, mean ± s.d.) is slightly lower than it is with allocentric correction (*Figure 3C*; 0.17 ± 0.11, mean ± s.d.). Excluding phase-independent neurons with fit score <0.3 (*Appendix 2—figure 1D*), we see a large population of phase-locking neurons and a small population of phase-precessing neurons (*Appendix 2—figure 1E*). If we use a high phase range cutoff of 120° for phase precession, we find that preferred phase predominantly decreases with progress through the grid field, which is the biologically observed direction. By decoding position from spike trains and averaging over theta cycles, we see a sawtooth pattern containing theta sequences (*Appendix 2—figure 1F*), though the pattern is less organized than it is with allocentric correction (*Figure 5B*).

During idle periods, traveling wavefronts produce replays (*Appendix 2—figure 1G* and *Appendix 2—video 1*). The speeds of theta sequences and replays are 1.1 ± 0.3 m/s and 7.0 ± 2.6 m/s respectively (mean ± s.d.; *Appendix 2—figure 1H*), which agree well with corresponding speeds in simulations with allocentric correction (0.9 ± 0.2 m/s and 7.0 ± 2.6 m/s), and which also agree well with experiments (*O'Neill et al., 2017*). As in *Figure 8A–C*, the lower velocity gain produces a bias in replay direction away from the current animal position (*Appendix 2—figure 1I,J*).

In summary, after removing allocentric corrections, our model maintains its features involving theta sequences and replays. It still produces some degree of phase precession, but for fewer neurons and larger precession ranges.

