## [Decision Letter]

**Acceptance summary:**

Continuous attractor networks have been used to simulate grid cells. This paper shows that the continuous attractors can be easily adapted to show other properties of the entorhinal-hippocampal network, namely phase precession, theta sequences during behavior and replay sequences during rest. This detailed and comprehensive modelling study shows that when the continuous attractor network is coupled with theta modulated inhibition, the attractor bumps expand and contract, leading to theta phase precession and theta sequences. Reduction of external drive leads to spontaneous replay like sequences. These results are of significant interest to both theoreticians as well as experimentalists interested in temporal phenomena in the grid cell network.

**Decision letter after peer review:**

Thank you for sending your article entitled "Replay as wavefronts and theta sequences as bump oscillations in a grid cell attractor network" for peer review at *eLife*. Your article is being evaluated by three peer reviewers, and the evaluation is being overseen by a Reviewing Editor and Laura Colgin as the Senior Editor.

Given the list of essential revisions, including major changes to the model and extensive new analyses, the editors and reviewers invite you to respond within the next two weeks with an action plan and timetable for the completion of the additional work. The action plan should contain a similar amount of detail as a traditional rebuttal letter. We plan to share your responses with the reviewers and then issue a binding recommendation.

Essential revisions:

1) The non-periodic boundary conditions and non-uniform driving inputs seem like rather odd design choices. This raises some concern that there may be unexplained reasons for these choices (that is, the model's behavior may depend upon these choices in ways that are not spelled out). The model should be re-implemented using periodic topology, either across the entire manuscript or in additional comparison figures.

2) Phase precession in this model may differ from experimentally observed phase precession in some fundamental ways. The Hafting et al., 2008, study shows that layer II MEC phase precession is highly stereotyped, with typical entry/exit phases of 222º/60º, corresponding to the total ~160º precession. Further, Hafting reported typical slopes of about -3º/cm for 30-70 cm grid spacings; "correcting" this value by mean([30,70])/120 for the larger ~120 cm spacing of the modeled grids (Figure 1E) suggests modeled phase slopes should be on the order of -1.25º/cm. But the authors show the slope ranging up to -30º/cm; this range seems clearly too high. It could be that the trajectory-dependent starting phase produces phase trajectories that yield essentially meaningless circular-linear regression slopes as possibly indicated by Figure 3B. Or, the largest bar in Figure 3D could be correctly showing this smaller slope (instead of phase locking as interpreted by the authors), and the interpretation of the larger slopes as precession was simply incorrect. Regardless, the authors must revise the phase precession mechanism and/or its interpretation to at least qualitatively recapitulate the limited range, grid-scale appropriate slopes, and stereotyped phase trajectories in published data.

3) It is unclear whether the 1D phase precession shown in the paper results are specific to a chosen trajectory through the 2D sheet, or whether these results generalize across all directions. The justification for the track choice to improve "interpretability" is weak given that the choice reduces the robustness of the results. The results should be directionally generalized, and the effects of multiple fields on the track (possibly using larger tracks commensurate with grid scale) should be demonstrated and interpreted. Furthermore, the authors need to demonstrate generalization across a range of speeds.

4) A parametric analysis of the conditions which give rise to phase precession and wavefront propagation is rather lacking here, and should be included.

5) Theta sequences: The O'Neill paper (Supplementary Figure 10E) shows the clear sawtooth pattern in which MEC sequences start from behind and end up ahead of the current position. The stalling/speeding dynamics are qualitatively different from both hippocampal observations, and the O'Neill MEC data. The authors should ensure the validity of theta sequence output with regards to (at the minimum) their stated benchmark.

Reviewer #1:

Continuous attractor networks have been used to simulate grid cells. This paper shows that the continuous attractors can be easily adapted to show other properties of the entorhinal-hippocampal network, namely phase precession, theta sequences during behavior and replay sequences during rest.

While the idea of getting these cellular properties out of a network that hasn't been specifically designed to generate these is very attractive, I suspect that some of these observations may arise specifically out of biologically unrealistic idiosyncrasies of the model.

1) "During locomotion, excitatory principal cells are driven to fire by various neocortical and thalamic inputs [Kerr et al., 2007]. Thus, their drive exceeds threshold at the center of the neural sheet; it decays towards the edges to avoid edge effects that disrupt continuous attractor dynamics [Burak and Fiete, 2009]." And "Once nucleated, the wavefront propagates freely through the outer region because inhibitory neurons, with low drive, are not activated by the wavefront. In contrast, wavefronts cannot propagate in the center region, because excitatory neurons there receive enough drive to spike vigorously and activate nearby inhibitory neurons."

During quiescent periods, the excitatory drive reduces, but still maintains its shape (Figure 5). The replay events are seen only at the periphery and not at the center of the sheet. I'm concerned that 1) This excludes center of the sheet from replays (which is not supported by any observed data to the best of my knowledge) and 2) The replays could simply be a side effect of the simulation architecture (which may not be replicated in reality – no one has demonstrated grid cells which maintain location specific firing during quiescent periods), and may not be seen if the boundary effects are countered by using a toroid instead of a sheet with gaussian excitatory drive peaking at the center of the sheet. The authors do discuss a possible enhanced model with landmark and border inputs sustaining their activity through the quiescent period (Discussion, fifth paragraph) that they claim can deal with this issue, but that is insufficient. The authors should implement the attractor as a toroid instead of as a sheet along with external drive not limited to a part of the network ("center" in case of the sheet) which can persist (at a lower level) during the quiescent period, rather than base a main result on the specific architecture that is unrealistic.

This change will also deal with the need for the external drive to be limited to the center during locomotion.

Theta phase precession and sequences as well as replay sequences arising out of such a model would be much more convincing.

2) "Unlike the 2D case in which grid cells have multiple firing fields, we choose parameters such that they have single fields to improve the interpretability of our results."

The authors need to show that their results are not unique to the selected direction, even if it’s just by showing that the results are at least qualitatively similar in other directions (even if the multiple vertices create redundancy). One possible concern is that the replays can "jump" between redundant vertices when a neuron with multiple fields on the track is active (i.e. instead of going in the order 1 2 3 N 4 5 6, redundant fields of neuron N can give rise to a sequence 1 2 3 N 25 26 27). This can degrade the replay.

3) Figure 3B, G. Theta phase precession shown here is unlike the observed precession, in the sense that there appear to be double/multiple precessing lines running within a field (this can be alternatively interpreted as precessing through >> 360°). This could be an artefact of the chosen refractory period, and the precession might disappear when using a shorter refractory period. A 40ms refractory period is rather long for grid cells. Does having a shorter refractory period drastically alter the results? Cells showing 10-15°/cm seem to have more realistic precession, but even that might get affected by having a shorter refractory period.

4)." Wavefront propagation involves dynamical processes that are fundamentally different from attractor bump motion, and thus leads to different decoded speeds and directions for replays and theta sequences."

How dependent is the wavefront propagation speed on the specific time constants used in the model? Were the time constants tweaked to achieve the desired speed?

Reviewer #2:

In this paper, Kang and DeWeese present a spiking model of the entorhinal grid cell network. The paper's main contributions are rooted in two novel ideas: 1) explaining phase precession as a consequence of theta-synchronized expansion and contraction of attractor bumps, and 2) showing that replay sequences can occur during propagating waves that emerge in regions of the network where excitatory drive is low. There are several issues that I think should be addressed in revision:

1) The idea that phase precession arises from expansion/contraction of attractor bumps is quite innovative, but I am concerned that under this mechanism, the phase at which grid cells fire during entry / exit from their fields may not fully accord with experimental findings. Two issues should be addressed here:

1a) What model parameters determine whether a particular grid cell has a fixed versus variable phase? Does this depend upon movement velocity (speed or direction), the neuron's position in the sheet, the level of excitatory drive, etc.? It would be helpful if the authors could perform a more systematic analysis of the drive / speed / direction regimes within which different phase precession behaviors occur. One of the paper's main strengths is its novel theory for phase precession, but without a clearer understanding of how this phase precession behavior arises from the network dynamics, it is difficult to understand what this theory entails, and what it might predict for biology.

1b) While explaining phase precession by expansion and contraction of the attractor bumps is quite innovative, I am concerned that phase precession arising from this mechanism may differ in a fundamental way from experimentally observed phase precession. Before explaining this concern, it is necessary to point out that the authors have adopted a convention for measuring theta phase that is opposite from what many experimental authors have previously used. Here, the authors define 180° as the phase at which grid cell firing rates are at their *minimum*. By contrast, classical phase precession papers by O'Keefe, the Mosers, and others defined 180° as the phase at which firing rates of place/grid cells are at their *maximum*. This reversal of convention must be carefully tracked when assessing whether the model agrees with evidence from prior papers. To avoid any confusion arising from this discrepancy, let me state my concern in a way that is independent of which phase convention is used. In experimental studies, place and grid cells usually show a specific preferred firing phase as the animal *enters* the spatial field, which is ~150-170° *later* than the phase at which the cell fires when the animal is in the center of its field. However, in the present model, it appears that this is not the case. For example, in Figure 3G, it appears that as the animal enters a grid cell's field, the cell prefers to fire at a phase which is a few degrees *earlier* than the phase at which the cell fires when the animal is in the center of its field. I don't think this is in good accordance with experimental findings for grid cells or place cells. This issue should be explicitly addressed in the paper.

2) The authors have chosen to build their network with non-periodic boundary conditions (rectangular sheets of neurons) and a radial gradient of excitatory drive which peaks in the center of the sheet. It is unclear how essential behaviors of the model (grid cells with diverse phase precession profiles, propagating waves) depend upon the non-uniform excitatory drive, or whether these behavior could also be obtained under uniform drive.

3) The influence (if any) of movement velocity upon phase precession is not clear. In a 2D attractor network, velocity has two components: speed and direction. Does the phase precession behavior of the model depend upon one or both of these velocity components? Concerning speed, it is not entirely clear whether simulation results were obtained using fixed or variable running speeds (for example, in Figure 4C, why is there a nonzero error bar for "actual" running speed? It does not appear from the trajectory plot in Figure 2A that there is much noise in the running speed during simulated track traversals). Concerning direction, was the 1D track simulated as some specific linear trajectory through the 2D grid sheet, and if so, what was the orientation alignment of the 1D trajectory on the 2D grid sheet? Would different alignments yield fundamentally different phase precession results?

4) If excitatory drive is the main determinant of wave propagation, then it would be nice if the authors could perform a more systematic analysis of the drive regimes in which wave propagation occurs. Can the entire sheet be switched into wave propagation behavior by applying an appropriately low level of uniform excitatory drive? Or is there something about the non-uniformity of the drive level that is essential for the wave propagation to occur?

Reviewer #3:

Kang and DeWeese present a detailed and comprehensive modeling study of theta-rhythmic modulation of the activity bump on a continuous attractor map to explain phase precession, theta sequences, and, by reducing subcortical input levels, the switch to the "irregular" state in which sharp-wave/ripple 'replay' reactivations are observed. These are well-studied hippocampal phenomena that may be crucial to the spatial, mnemonic, and decision-making roles of the hippocampus; the manuscript addresses how they may arise in medial entorhinal (MEC) grid cell networks, which is a critical question for the field. The attractor-based mechanism is interesting and potentially contributes a compelling unification of these temporal phenomena; however, there are several gaps in logic and disconnections from the literature in the way the results are currently presented.

• The claims regarding phase precession are largely undercut by the low correlations in Figure 3C and zero-dominated histogram in Figure 3D showing that most of the simulated grid cells do not precess. The authors note that additional features may increase the correlations, but the bias of the distribution toward 0 is the problem, not the average value. Hafting et al., 2008, showed that layer II principal cells all generally precess; while the authors mention that phase locking is observed in layer III (cf. the Hafting paper) as justification, their model is not a layer III model. If the bump modulation is the shared cause of precession and sequences, then the two effects should be equivalently strong in the model.

• Additionally, arguments regarding hippocampal phase precession have related to whether it is a cause or effect of temporal compression during theta sequences, or even whether it is epiphenomenal. In MEC, if both effects are generated by the same underlying mechanism as in this model, then the critical question concerns whether phase precession has any function in this setting. Some theories posit that precession helps construct sequences, but others claim that it is simply a by-product of temporal compression (e.g., G. Dragoi and G. Buzsáki. Neuron, 50(1):145-157, 2006.; G. Dragoi. Front Syst Neurosci, 7:46, 2013). It's not clear what the authors are claiming in this manuscript, but the results argue for precession as epiphenomenon because it is unnecessary to either sequences or replay, and the starting phase at the edge of a grid field is apparently random, indicating that it plays no role in the spatiotemporal ordering of activity. The description of the origin of precession in the model (subsection “Theta phase precession arises from oscillations in attractor bump size”, third paragraph) emphasizes that it results from transient changes in spread of activity, instead of external input, network, or cell-assembly modulation as in hippocampal models. The authors should broadly reconsider and clarify the theoretical context and implications of their model design and results regarding precession and theta sequences.

• The wavefront replay mechanism suffers from a similar causative discrepancy in which the replay sequences do not have any particular relationship with recently expressed sequences. Experiments (e.g., Drieu, Todorova, and Zugaro, 2018) are beginning to corroborate *earlier* theories (such as Dragoi, 2013, mentioned above) that online sequences drive plasticity necessary for subsequent replay. However, grid-cell weights in the model are hard-wired, thus precluding any direct contribution to the learning or memory consolidation roles that may be central to the replay phenomenon in general. The Discussion argues that hard-wired replay might play an initiation role for hippocampal replays, but that incorrectly assumes grid cells directly activate place cells (as in the old grid-to-place models) and ignores sharp-wave initiation studies implicating dentate gyrus, CA2, and/or ascending inputs (Sasaki, et al., 2018; Oliva, et al., 2016; Angulo-Garcia, et al., 2018). The potential roles of grid cell wavefronts may be more limited than discussed, which should be addressed.

• The model is based on the aperiodic network with feedforward Gaussian envelope function described by Burak and Fiete, 2009, but they also described a periodic solution. The periodic solution has problems with rotation, but some of the results (especially the generation of replays) are affected by or depend on the topology. The authors should clarify the reliance of their results on network topology and whether other formulations may also support their conclusions.

---

## [Author Response]

Essential revisions:1) The non-periodic boundary conditions and non-uniform driving inputs seem like rather odd design choices. This raises some concern that there may be unexplained reasons for these choices (that is, the model's behavior may depend upon these choices in ways that are not spelled out). The model should be re-implemented using periodic topology, either across the entire manuscript or in additional comparison figures.

This is a good point. Following the Editor and reviewers’ suggestion, we have added comparison figures featuring a model with periodic topology and uniform drive (Figure 8D–G). It reproduces all essential replay and theta-related behaviors of the non-periodic implementation (Figure 8—figure supplement 2 and Figure 8—figure supplement 3). As in the non-periodic case, it transitions between attractor bumps arranged in a triangular grid and traveling wavefronts in response to changes in input drive. The attractor bumps are capable of path-integration during runs.

During quiescent periods, traveling wavefronts propagate across the entire network. This erases information about the animal’s position, so grid fields would not be consistent from lap to lap. To maintain positional information, we have added a simple implementation of brief allocentric input from border, boundary vector, or landmark cells (Ocko et al., 2018; Keinath et al., 2018; and others). This input is active only for 0.5 s between idle periods and runs to nucleate the activity bumps in their appropriate conditions. Inputs to the model are strictly uniform during the 1.5 s-long runs and the 2.0 s-long periods of quiescence that contain traveling wavefronts.

For most of the paper, we still use non-periodic boundary conditions, which requires non-uniform driving inputs (Burak and Fiete, 2009). Although periodic boundary conditions are useful to represent a large area of cortex in which edges are deemed irrelevant, we find that non-periodic boundary conditions are more biologically feasible. Periodic boundary conditions obeying a toroidal topology require a precise configuration of long-range connections that we believe is unlikely to exist in the brain. On the other hand, continuous attractor networks with non-periodic boundary conditions can arise from anatomically local connections, whose precise symmetry can be learned through synaptic plasticity rules (Widlowski and Fiete, 2014).

2) Phase precession in this model may differ from experimentally observed phase precession in some fundamental ways. The Hafting et al., 2008, study shows that layer II MEC phase precession is highly stereotyped, with typical entry/exit phases of 222º/60º, corresponding to the total ~160º precession. Further, Hafting reported typical slopes of about -3º/cm for 30-70 cm grid spacings; "correcting" this value by mean([30,70])/120 for the larger ~120 cm spacing of the modeled grids (Figure 1E) suggests modeled phase slopes should be on the order of -1.25º/cm. But the authors show the slope ranging up to -30º/cm; this range seems clearly too high. It could be that the trajectory-dependent starting phase produces phase trajectories that yield essentially meaningless circular-linear regression slopes as possibly indicated by Figure 3B. Or, the largest bar in Figure 3D could be correctly showing this smaller slope (instead of phase locking as interpreted by the authors), and the interpretation of the larger slopes as precession was simply incorrect. Regardless, the authors must revise the phase precession mechanism and/or its interpretation to at least qualitatively recapitulate the limited range, grid-scale appropriate slopes, and stereotyped phase trajectories in published data.

We agree with the Editor’s and reviewers’ suggestions and have revised our manuscript accordingly, as we will now describe. We will structure our response by successively addressing stereotyped phase trajectories, grid-scale appropriate slopes, and limited range, as listed in the last sentence of the Concern.

First, we discuss stereotyped phase trajectories. Our new implementation of brief allocentric corrections has increased the number of phase-precessing neurons in our model. However, we do not believe that the experimental evidence requires our model to exhibit the same degree of precession across all neurons. These two points are elaborated in turn below.

a) Brief allocentric corrections have made phase behavior more stereotyped in our model. Our original results were based on simulated data from only 12 laps along the track, 6 in each direction, which provided many fewer spikes per neuron than were acquired in phase precession experiments. Our lap count was limited by attractor drift, which arises from noise accumulation and destabilizes grid fields. Allocentric input is believed to correct errors in path-integration (Hardcastle et al., 2015; Ocko et al., 2018; Keinath et al., 2018; and others; see also Essential revision1) and allows us to run simulations for many more laps, providing more spikes per neuron with more stable firing fields. It also allows us to use stronger velocity input, which in our experience produces more attractor drift because the attractor bumps move farther along the neural sheet during each lap.

Compared to our original simulations, more neurons exhibit phase precession between 60º and 360º and a smaller proportion of neurons exhibit phase-position correlation close to 0 (Figure 3). Although correlation values are still less than they are in Supplementary Figure 10C of O’Neill et al., 2017, the peak in the distribution is no longer at 0, as observed in experiment. In the Discussion, we mention how these correlation values can be further increased with additional simulation features that may reduce the number of phase-independent neurons that fire throughout the theta cycle.

b) Although our neurons do not all precess in a stereotyped fashion, this heterogeneity matches the variety of phase behaviors observed in superficial MEC layers. We believe that at most, the experimental literature indicates that only a subpopulation of neurons in superficial MEC layers exhibits highly stereotyped phase precession. Hafting et al., 2008 report that LII neurons consistently precess, whereas LIII neurons exhibit phase locking instead. However, LII pyramidal cells precess less than LII stellate cells do (Ebbesen et al., 2016; Reifenstein et al., 2016) and even less than LIII neurons do at the single-lap level (Ebbesen et al., 2016).

It is not certain that our grid cells should correspond to one subpopulation of MEC neurons in particular. Grid cells in continuous attractor models, like our model, increase their activity when the animal moves along a preferred direction and can thus be considered grid × head direction conjunctive cells (Zilli, 2012). These conjunctive cells are more common in LIII and deeper layers (Sargolini et al., 2006). It is true that other features of continuous attractor models, such as recurrent inhibition (Couey et al., 2013), may be better established in LII. Additionally, some researchers do propose that continuous attractor models describe LII neurons with directional inputs from deeper MEC layers (Bush and Burgess, 2014; and others). Nevertheless, we find it too restrictive to claim that our model must describe only MEC LII neurons (and potentially only its strongly precessing stellate cells). The O’Neill et al., 2017 results that our model addresses were obtained from superficial MEC layers without distinguishing between LII and LIII; they contain a mixture of phase behaviors.

Next, we discuss grid-scale appropriate slopes. In short, to properly make the important experimental comparisons mentioned by the Editor and reviewers, we have decided to convert our slopes to units of degrees per field instead of degrees per cm. In our manuscript, we focused on matching theta behavior and replay as measured by O’Neill et al., 2017. Their grid cells have scale ~150 cm, and their example of phase precession exhibits slope ~10º/cm (Supplementary Figure 10A in O’Neill et al., 2017). It is true that Hafting et al., 2008 provides a more comprehensive investigation of phase precession, and both their grid scales (30–70 cm) and phase precession slopes (~3º/cm) are much smaller. Other papers report precession slopes in units of degrees per second (Ebbesen et al., 2016; Reifenstein et al., 2016). In light of these differences in slope values even among experimental reports and their dependence on grid scale, we find that a more consistent metric for precession is its phase range, which equals the absolute value of precession slope in units of degrees per field. We will discuss experimental comparisons of precession range in the next paragraph.

Finally, we discuss the range of phase precession. The Editor and reviewers point out that our previous results exhibit large phase precession slopes of magnitude ~30º/cm, which corresponds to a precession range of greater than 360º. Our new simulations incorporate brief allocentric input and better align with precession ranges reported experimentally, which are less than 360º. We present two analyses of precession range in our revised manuscript:

a) Figure 3E shows our analysis of phase precession slope in units of field size, and its absolute value is the precession range. We agree with the Editor’s suggestion that some slope values may be meaningless due to poor fit. To address this issue, we modified our slope analysis by only including neurons whose fit score, which is the mean resultant length (Kempter et al., 2012), exceeds a certain value. In Figure 3E, neurons still preferentially show negative slopes for rightward runs and positive slopes for leftward runs, which corresponds to decreasing phase as the animal moves through the field. Moreover, many neurons responsible for this bias, corresponding to regions in which the “left” and “right” histograms do not overlap, precess with range 60º–360º. This compares favorably with these median precession ranges experimentally measured in MEC LII: ~160º in cells of unspecified type (Hafting et al., 2008), ~170º and ~110º in stellate and pyramidal cells respectively (Reifenstein et al., 2016), and ~130º and ~50º in stellate and pyramidal cells respectively (Ebbesen et al., 2016). Our simulations still produce many cells with precession range close to 0º, which may correspond to MEC LIII cells observed to lack strong precession (Hafting et al., 2008).

b) Following Hafting et al., 2008, we present in Figure 3F spike density plots averaged over different subgroups of neurons. The rightmost density plot corroborates our results in Figure 3E: for neurons with high fit scores and large phase ranges >60º, the preferred phase decreases as the animal moves through grid fields. This preferred phase is ~360º (or equivalently 0º) at the beginning of the field, and magnitude of its decrease is ~75º. For neurons with small phase ranges <60º, neurons maintain a constant phase preference of ~360º. The former population appears to be consistent with LII cells and the latter with LIII cells.

In summary, our implementation of brief allocentric corrections leads to more phase-precessing neurons whose precession range lies within the experimentally reported span for LII neurons. In the Discussion, we cite experimental measurements to explain why we do not believe our model should necessarily produce only grid cells with stereotyped precession.

3) It is unclear whether the 1D phase precession shown in the paper results are specific to a chosen trajectory through the 2D sheet, or whether these results generalize across all directions. The justification for the track choice to improve "interpretability" is weak given that the choice reduces the robustness of the results. The results should be directionally generalized, and the effects of multiple fields on the track (possibly using larger tracks commensurate with grid scale) should be demonstrated and interpreted. Furthermore, the authors need to demonstrate generalization across a range of speeds.

This is a good point. We have now generalized our results to a variety of track directions relative to the neural sheet and across a range of speeds (Figure 3—figure supplement 6). We have also compared phase behavior for simulations exhibiting single fields and multiple fields (Figure 3—figure supplement 5). We have found that precession range and correlation magnitude increase with animal speed, as observed experimentally (Jeewajee et al., 2014). However, track orientation and the number of grid fields do not affect phase behavior.

4) A parametric analysis of the conditions which give rise to phase precession and wavefront propagation is rather lacking here, and should be included.

We agree that a parametric analysis is a good idea. We have now measured various phase precession, theta sequence, and replay properties across a wide range of parameters. The results are shown in Figure 3—figure supplement 6, Figure 3—figure supplement 7, Figure 5—figure supplement 3, and Figure 7—figure supplement 4. Across all parameter values, theta sequence speed remains roughly 2 times the actual speed, and replay speed remains roughly 14 times the actual speed. These values are consistent with our original findings and with experimental measurements (O’Neill et al., 2017).

5) Theta sequences: The O'Neill paper (Supplementary Figure 10E) shows the clear sawtooth pattern in which MEC sequences start from behind and end up ahead of the current position. The stalling/speeding dynamics are qualitatively different from both hippocampal observations, and the O'Neill MEC data. The authors should ensure the validity of theta sequence output with regards to (at the minimum) their stated benchmark.

The Editor and the reviewers make a good point; in our new simulations with brief allocentric corrections (see also Essential revision1), theta sequence plots now exhibit a prominent sawtooth pattern (Figure 5B). These sequences start behind the animal’s actual position and end ahead of it (Figure 5D), a feature that arises from activity bump dynamics (Figure 6D). Allocentric corrections allow for combination of more simulation laps, more stable firing fields, and stronger velocity input that cause this sawtooth pattern to emerge.

Reviewer #1:Continuous attractor networks have been used to simulate grid cells. This paper shows that the continuous attractors can be easily adapted to show other properties of the entorhinal-hippocampal network, namely phase precession, theta sequences during behavior and replay sequences during rest.While the idea of getting these cellular properties out of a network that hasn't been specifically designed to generate these is very attractive, I suspect that some of these observations may arise specifically out of biologically unrealistic idiosyncrasies of the model.1) "During locomotion, excitatory principal cells are driven to fire by various neocortical and thalamic inputs [Kerr et al., 2007]. Thus, their drive exceeds threshold at the center of the neural sheet; it decays towards the edges to avoid edge effects that disrupt continuous attractor dynamics [Burak and Fiete, 2009]." And "Once nucleated, the wavefront propagates freely through the outer region because inhibitory neurons, with low drive, are not activated by the wavefront. In contrast, wavefronts cannot propagate in the center region, because excitatory neurons there receive enough drive to spike vigorously and activate nearby inhibitory neurons."During quiescent periods, the excitatory drive reduces, but still maintains its shape (Figure 5). The replay events are seen only at the periphery and not at the center of the sheet. I'm concerned that 1) This excludes center of the sheet from replays (which is not supported by any observed data to the best of my knowledge) and 2) The replays could simply be a side effect of the simulation architecture (which may not be replicated in reality – no one has demonstrated grid cells which maintain location specific firing during quiescent periods), and may not be seen if the boundary effects are countered by using a toroid instead of a sheet with gaussian excitatory drive peaking at the center of the sheet. The authors do discuss a possible enhanced model with landmark and border inputs sustaining their activity through the quiescent period (Discussion, fifth paragraph) that they claim can deal with this issue, but that is insufficient. The authors should implement the attractor as a toroid instead of as a sheet along with external drive not limited to a part of the network ("center" in case of the sheet) which can persist (at a lower level) during the quiescent period, rather than base a main result on the specific architecture that is unrealistic.This change will also deal with the need for the external drive to be limited to the center during locomotion.Theta phase precession and sequences as well as replay sequences arising out of such a model would be much more convincing.

The reviewer raises an interesting point here. As we have described in our response to Essential revision1, we have now implemented periodic boundary conditions with uniform drive, and our simulations show attractor bumps and traveling wavefronts during runs and idle periods, respectively (Figure 8). These grid cells no longer maintain location-specific firing during quiescent periods. However, we do not believe there is convincing experimental evidence for or against location-specific firing during quiescent periods. Few papers report the properties of grid cells during quiescence; grid cells during this state do have much lower firing rates, but there appears to be some residual activity outside of highly synchronized events (Figure 2A in O’Neill et al., 2017). To our knowledge, it is unknown whether this activity maintains a representation of an animal’s location. Nevertheless, we expect our results to hold in either case; the animal’s spatial representation can be either maintained by an active region of the grid network (whether in the center or close to an edge) or reset by brief allocentric inputs.

2) "Unlike the 2D case in which grid cells have multiple firing fields, we choose parameters such that they have single fields to improve the interpretability of our results."The authors need to show that their results are not unique to the selected direction, even if it’s just by showing that the results are at least qualitatively similar in other directions (even if the multiple vertices create redundancy). One possible concern is that the replays can "jump" between redundant vertices when a neuron with multiple fields on the track is active (i.e. instead of going in the order 1 2 3 N 4 5 6, redundant fields of neuron N can give rise to a sequence 1 2 3 N 25 26 27). This can degrade the replay.

This is a good idea. As described in our response to Essential revision3, we agree that running simulations with varied track orientations and multiple grid fields would test the robustness of our model. We have included these results throughout our paper. Changing track orientation does not affect phase behavior (Figure 3—figure supplement 6). Simulations exhibiting one or two grid fields exhibit the same phase behavior and theta sequence speed (Figure 3—figure supplement 5, Figure 5—figure supplement 3). During replays, simulations with two grid fields can have their decoded probability split between multiple track positions that form parallel lines (Figure 7—figure supplement 3). Their detected speed is slightly lower than the speed of replays among grid cells with one field, but the difference in detection methods that accounts for split probability distributions may contribute to this difference (Figure 7—figure supplement 4).

3) Figure 3B, G. Theta phase precession shown here is unlike the observed precession, in the sense that there appear to be double/multiple precessing lines running within a field (this can be alternatively interpreted as precessing through >> 360°). This could be an artefact of the chosen refractory period, and the precession might disappear when using a shorter refractory period. A 40ms refractory period is rather long for grid cells. Does having a shorter refractory period drastically alter the results? Cells showing 10-15°/cm seem to have more realistic precession, but even that might get affected by having a shorter refractory period.

As described in our response to Essential revision2, our simulations with brief allocentric inputs, which allow us to obtain more spikes per neuron with more stable firing fields, demonstrate more neurons whose range of precession is less than 360º. We do not implement a refractory period in our main simulations; we have only considered conceptual models in Figure 4 that use a refractory period. In our main model, we do use a 40 ms membrane time constant for excitatory neurons. As described in our response to Essential revision4, we have characterized how the behavior of the model depends on the membrane time constants and other parameter values (Figure 3—figure supplement 6 and Figure 3—figure supplement 7). Phase precession properties do not significantly vary with membrane time constants over the range we tested.

4) " Wavefront propagation involves dynamical processes that are fundamentally different from attractor bump motion, and thus leads to different decoded speeds and directions for replays and theta sequences."How dependent is the wavefront propagation speed on the specific time constants used in the model? Were the time constants tweaked to achieve the desired speed?

Wavefronts, which arise directly from activity propagation from one excitatory neuron to the next, generally travel much faster than activity bumps, whose motion arises from smaller imbalances among excitatory populations with different preferred directions. Thus, replays are generally much faster than the animal trajectory. Without fine tuning, our choice of parameters makes replays ~14 times faster. As described in our response to Essential revision4, we have characterized how replay speed depends on the values of time constants and other parameters (Figure 7—figure supplement 4). In general, replays are 8–20 times faster than the speed of the animal trajectory, which agrees well with experimental measurements of a ~9–15-fold speed-up (O’Neill et al., 2017).

Reviewer #2:In this paper, Kang and DeWeese present a spiking model of the entorhinal grid cell network. The paper's main contributions are rooted in two novel ideas: 1) explaining phase precession as a consequence of theta-synchronized expansion and contraction of attractor bumps, and 2) showing that replay sequences can occur during propagating waves that emerge in regions of the network where excitatory drive is low. There are several issues that I think should be addressed in revision:1) The idea that phase precession arises from expansion/contraction of attractor bumps is quite innovative, but I am concerned that under this mechanism, the phase at which grid cells fire during entry / exit from their fields may not fully accord with experimental findings. Two issues should be addressed here:1a) What model parameters determine whether a particular grid cell has a fixed versus variable phase? Does this depend upon movement velocity (speed or direction), the neuron's position in the sheet, the level of excitatory drive, etc.? It would be helpful if the authors could perform a more systematic analysis of the drive / speed / direction regimes within which different phase precession behaviors occur. One of the paper's main strengths is its novel theory for phase precession, but without a clearer understanding of how this phase precession behavior arises from the network dynamics, it is difficult to understand what this theory entails, and what it might predict for biology.

We appreciate the reviewer’s comment that our model is based on an innovative mechanism. The reviewer makes a good suggestion that we determine differences between grid cells that exhibit phase locking and phase precession, and now we have performed a comprehensive analysis. To constrain spiking to preferred theta phases, both phase-locking and precessing neurons require low excitatory drive and tend to be located towards the edge of the neural sheet where drive is lower (Figure 3—figure supplement 6). In particular, phase-locking neurons must have spikes constrained to only the most permissive part of the theta cycle; thus, they have lower firing rates than phase-precessing neurons (Figure 3—figure supplement 4). Two factors contribute to lower firing rates: phase-locking neurons are slightly less prevalent towards the center of attractor bumps (Figure 3—figure supplement 4) and more prevalent in excitatory populations whose preferred direction is perpendicular to the track orientation (Figure 3—figure supplement 5). This relationship between precession range and firing rate was observed experimentally by Jeewajee et al., 2014, who reported that grid cells with more spikes have steeper precession slopes.

Furthermore, with a new simplified model of phase precession, we demonstrate that the level of neural activity can directly influence phase behavior (Figure 4E, F and Figure 4—figure supplement 1). In this model, we determine the time-varying shape of the average attractor bump and rescale its activity to different levels. We find that low, medium, and high activity levels increase the prevalence of phase-locking, precessing, and independent neurons respectively.

Phase precession properties are similar between populations whose preferred direction is either aligned with or perpendicular to the track orientation (Figure 3—figure supplement 5), which is related to omnidirectional precession observed experimentally (Climer et al., 2013). Precession is also independent of track orientation (see also Essential revision3), but its range increases with animal speed (Figure 3—figure supplement 6), as observed experimentally (Jeewajee et al., 2014). In our parametric analysis (see also Essential revision4), we report how phase precession properties vary in a complex fashion with simulation parameters (Figure 3—figure supplement 6 and Figure 3—figure supplement 7).

1b) While explaining phase precession by expansion and contraction of the attractor bumps is quite innovative, I am concerned that phase precession arising from this mechanism may differ in a fundamental way from experimentally observed phase precession. Before explaining this concern, it is necessary to point out that the authors have adopted a convention for measuring theta phase that is opposite from what many experimental authors have previously used. Here, the authors define 180° as the phase at which grid cell firing rates are at their minimum. By contrast, classical phase precession papers by O'Keefe, the Mosers, and others defined 180° as the phase at which firing rates of place/grid cells are at their maximum. This reversal of convention must be carefully tracked when assessing whether the model agrees with evidence from prior papers. To avoid any confusion arising from this discrepancy, let me state my concern in a way that is independent of which phase convention is used. In experimental studies, place and grid cells usually show a specific preferred firing phase as the animal enters the spatial field, which is ~150-170° later than the phase at which the cell fires when the animal is in the center of its field. However, in the present model, it appears that this is not the case. For example, in Figure 3G, it appears that as the animal enters a grid cell's field, the cell prefers to fire at a phase which is a few degrees earlier than the phase at which the cell fires when the animal is in the center of its field. I don't think this is in good accordance with experimental findings for grid cells or place cells. This issue should be explicitly addressed in the paper.

As described in our response to Essential revision 2, our new simulations with brief allocentric corrections demonstrate more neurons whose range of precession is less than 360º. These neurons have a preferred firing phase that is higher (later) at the beginning of a grid field than at its center.

Figure 4A–C of our revision (corresponding to Figure 3G in our first submission) presents a simplified conceptual model that readily explains two features of phase precession in our model: preferred phases at the beginning of a grid field are close to 0º/360º (under our convention) and preferred phases tend to decrease, not increase, through the field. The simplified conceptual model cannot explain why the range of precession is less than 360º. We have made this point clearer in our revision.

As for the convention for theta phase, we agree that the presence of opposite definitions is confusing. Although classic phase precession papers identify 180º with maximum grid cell activity, other papers, including O’Neill et al., 2017 and Feng et al., 2015, identify 180º with minimum activity. Since our focus is to capture the results of O’Neill et al., 2017, we use the latter convention.

2) The authors have chosen to build their network with non-periodic boundary conditions (rectangular sheets of neurons) and a radial gradient of excitatory drive which peaks in the center of the sheet. It is unclear how essential behaviors of the model (grid cells with diverse phase precession profiles, propagating waves) depend upon the non-uniform excitatory drive, or whether these behavior could also be obtained under uniform drive.

As described in our response to Essential revisions1, our simulations with periodic boundary conditions and uniform drive show attractor bumps and traveling wavefronts across the entire neural sheet during runs and idle periods, respectively (Figure 8). This implementation can produce all the essential behaviors found in our original model (Figure 8—figure supplement 2 and Figure 8—figure supplement 3).

3) The influence (if any) of movement velocity upon phase precession is not clear. In a 2D attractor network, velocity has two components: speed and direction. Does the phase precession behavior of the model depend upon one or both of these velocity components? Concerning speed, it is not entirely clear whether simulation results were obtained using fixed or variable running speeds (for example, in Figure 4C, why is there a nonzero error bar for "actual" running speed? It does not appear from the trajectory plot in Figure 2A that there is much noise in the running speed during simulated track traversals). Concerning direction, was the 1D track simulated as some specific linear trajectory through the 2D grid sheet, and if so, what was the orientation alignment of the 1D trajectory on the 2D grid sheet? Would different alignments yield fundamentally different phase precession results?

As described in our response to Essential revision3, we agree that running simulations with varied track orientations would test the robustness of our model. Yes, the 1D track corresponds to a fixed orientation of 36º on the neural sheet (Appendix 1), chosen to avoid cardinal directions, which are preferred by our neurons. We have now tested different orientations 0º, 22.5º, and 45º, and found no dependence between orientation and phase precession properties (Figure 3—figure supplement 6). We note that we do not fix the orientation of the grid-like activity pattern on the neural sheet; it varies from simulation to simulation.

We do include variations in velocity in our simulated trajectory such that the speed varies between 0.4 and 0.6 m/s (Appendix 1). As part of our parametric analysis (Essential revision4), we have also tested different mean running speeds of 0.3 and 0.75 m/s. The range of phase precession increases with animal speed (Figure 3—figure supplement 6), as observed experimentally (Jeewajee et al., 2014).

4) If excitatory drive is the main determinant of wave propagation, then it would be nice if the authors could perform a more systematic analysis of the drive regimes in which wave propagation occurs. Can the entire sheet be switched into wave propagation behavior by applying an appropriately low level of uniform excitatory drive? Or is there something about the non-uniformity of the drive level that is essential for the wave propagation to occur?

Yes, simulations with periodic boundary conditions and uniform drive demonstrate wavefronts that propagate throughout the entire neural sheet (Essential revision 1; Figure 8). As part of our parametric analysis (Essential revision 4), we found that replays are roughly independent of excitatory drive; their number is reduced with higher inhibitory drive, since inhibition disrupts wavefront propagation (Figure 7—figure supplement 4).

Reviewer #3:Kang and DeWeese present a detailed and comprehensive modeling study of theta-rhythmic modulation of the activity bump on a continuous attractor map to explain phase precession, theta sequences, and, by reducing subcortical input levels, the switch to the "irregular" state in which sharp-wave/ripple 'replay' reactivations are observed. These are well-studied hippocampal phenomena that may be crucial to the spatial, mnemonic, and decision-making roles of the hippocampus; the manuscript addresses how they may arise in medial entorhinal (MEC) grid cell networks, which is a critical question for the field. The attractor-based mechanism is interesting and potentially contributes a compelling unification of these temporal phenomena; however, there are several gaps in logic and disconnections from the literature in the way the results are currently presented.

We thank the reviewer for stating that we are addressing a critical question for the field and that our proposed mechanism is interesting and potentially contributes a compelling unification of temporal phenomena. We have made many changes to our manuscript that we feel provide more connections to the literature and better logical flow.

• The claims regarding phase precession are largely undercut by the low correlations in Figure 3C and zero-dominated histogram in Figure 3D showing that most of the simulated grid cells do not precess. The authors note that additional features may increase the correlations, but the bias of the distribution toward 0 is the problem, not the average value. Hafting et al., 2008, showed that layer II principal cells all generally precess; while the authors mention that phase locking is observed in layer III (cf. the Hafting paper) as justification, their model is not a layer III model. If the bump modulation is the shared cause of precession and sequences, then the two effects should be equivalently strong in the model.

The Reviewer raises a subtle and important point. As described in our response to Essential revision2, we believe our model does not necessarily describe only neurons with stereotyped phase precession. Our neurons with grid-like firing maps and directional preferences can be considered conjunctive cells, which are more common in MEC LIII than in LII (Sargolini et al., 2006). These LIII cells exhibit phase locking more than phase precession (Hafting et al., 2008). Moreover, Ebbesen et al., 2016 found that the range and slope of phase precession in LII pyramidal cells is even less than those of LIII cells. We do not exclude the possibility that our model may capture LII neurons whose directional selectivity is influenced by conjunctive cells from deeper MEC layers (as suggested by Bush & Burgess, 2014; and others), but we do not agree that biological plausibility requires most of our cells to precess. We thank Reviewer 3 for pointing out that we did not fully discuss this in our manuscript, and now our justification is provided in the Discussion.

Moreover, incorporating brief allocentric corrections leads to a shift in the peak of our phase-position correlation distribution away from 0 (Figure 3C). In the Discussion, we suggest other biophysical features that may be added to our model to increase its correlation further.

• Additionally, arguments regarding hippocampal phase precession have related to whether it is a cause or effect of temporal compression during theta sequences, or even whether it is epiphenomenal. In MEC, if both effects are generated by the same underlying mechanism as in this model, then the critical question concerns whether phase precession has any function in this setting. Some theories posit that precession helps construct sequences, but others claim that it is simply a by-product of temporal compression (e.g., G. Dragoi and G. Buzsáki. Neuron, 50(1):145-157, 2006.; G. Dragoi. Front Syst Neurosci, 7:46, 2013). It's not clear what the authors are claiming in this manuscript, but the results argue for precession as epiphenomenon because it is unnecessary to either sequences or replay, and the starting phase at the edge of a grid field is apparently random, indicating that it plays no role in the spatiotemporal ordering of activity. The description of the origin of precession in the model (subsection “Theta phase precession arises from oscillations in attractor bump size”, third paragraph) emphasizes that it results from transient changes in spread of activity, instead of external input, network, or cell-assembly modulation as in hippocampal models. The authors should broadly reconsider and clarify the theoretical context and implications of their model design and results regarding precession and theta sequences.

The reviewer raises important questions regarding the interpretation of both published data and our modeling results. While our work focuses on the mechanistic origins of phase precession and theta sequences, we appreciate the suggestion to further consider their relationship to each other and to broader cognitive theories. Accordingly, we have expanded upon our comments on this topic in our Discussion.

As reviewer 3 mentions, in hippocampus, some work indicates that phase precession may appear before theta sequences (Feng et al., 2015), which is consistent with the possibility that phase precession gives rise to theta sequences. Our work suggests that such a dependence is not necessary in grid cells and that both can arise from the same underlying cause—namely, oscillations in inhibitory drive. (Similarly, Navratilova et al., 2012, also suggest that both phenomena are consequences of a single factor, after-spike depolarization, in a grid cell model.) Note that this is not inconsistent with a difference in the onset times of these two phenomena in MEC, which has not been observed to our knowledge but would be analogous to what Feng et al., 2015 report in hippocampus. In our model, phase precession arises from oscillations in attractor bump size and theta sequences arise from oscillations in bump velocity. Oscillating inhibitory drive causes both types of bump oscillations, but their relative prominence may differ from one parameter regime to another (Figure 6—figure supplement 1), which may cause phase precession to be detected without theta sequences, or vice versa, at certain times.

One possible implication of our model is that these theta-related phenomena in MEC, which depend only on intrinsic circuitry and medial septum inputs in our model, may drive the corresponding phenomena in hippocampus, where such particular intrinsic circuitry may not exist. In support of this idea are reports that grid cell phase precession occurs independently of hippocampus (Hafting et al., 2008), but place cell phase precession and theta sequences require the MEC (Schlesiger et al., 2015). Plasticity in the hippocampus can then help the animal leverage these phenomena during movement, like replays during quiescence, to pursue cognitive goals. In this sense, phase precession would not be an epiphenomenon that serves no purpose. For example, phase precession may enable phase coding of position, which allows more accurate assignment of rewards, and theta sequences may help the animal decide between possible trajectories (Kay et al., 2019) based on previous experience. However, our results hold independent of whether these ideas about the relationship between MEC and hippocampus are borne out by future experiments.

Finally, we note that spike phases at the beginning of grid fields are consistently close to 0º/360º in our model, and not random (Figure 3F), which aligns well with experimental data (Hafting et al., 2008; O’Neill et al., 2017). The simplified conceptual model presented in Figure 4 of our manuscript provides an intuitive explanation for this feature. We acknowledge the reviewer for the opportunity to emphasize this feature in our revision.

• The wavefront replay mechanism suffers from a similar causative discrepancy in which the replay sequences do not have any particular relationship with recently expressed sequences. Experiments (e.g., Drieu, Todorova and Zugaro, 2018) are beginning to corroborate earlier theories (such as Dragoi, 2013, mentioned above) that online sequences drive plasticity necessary for subsequent replay. However, grid-cell weights in the model are hard-wired, thus precluding any direct contribution to the learning or memory consolidation roles that may be central to the replay phenomenon in general. The Discussion argues that hard-wired replay might play an initiation role for hippocampal replays, but that incorrectly assumes grid cells directly activate place cells (as in the old grid-to-place models) and ignores sharp-wave initiation studies implicating dentate gyrus, CA2, and/or ascending inputs (Sasaki, et al., 2018; Oliva, et al., 2016; Angulo-Garcia et al., 2018). The potential roles of grid cell wavefronts may be more limited than discussed, which should be addressed.

We acknowledge that our discussion of a potential role for grid cell replay, the driving of hippocampal replay, was limited, and we thank the reviewer for bringing this point and the broader literature on sharp-wave initiation to our attention. There can be advantages to having replays driven by intrinsically generated sequences, like a central pattern generator, without synaptic plasticity. These replays would be available immediately and maintain an unbiased spatial distribution. However, it is completely possible that there are multiple sites contributing to sequence initiation, including the ones mentioned by reviewer 3. Although the main results of our paper do not depend on the cognitive roles assigned to grid cell replays, we have revised our Discussion to emphasize the broader context of other potential drivers of hippocampal replay.

The potential role for grid cell replay to drive hippocampal replay does assume that grid cells activate place cells. This activation does not have to be direct; the flow of activity from grid cells in MEC LII to classical place cells in CA1, for example, can travel through dentate gyrus and CA3, although grid cells or conjunctive cells in MEC LIII may also directly activate CA1 through the temporoammonic pathway (Brun et al., 2008). The idea that grid cells influence place cells is supported by classical studies (see Bush et al., 2014 for a review) and more recent work (Hales et al., 2014; Kanter et al., 2017; Mallory et al., 2018). However, we are not asserting that grid cells are the sole drivers of place cell activity (Bush et al., 2014), and recent work also suggests that grid cells can be influenced by (Ólafsdóttir et al., 2016; Boccara et al., 2019; Butler et al., 2019) or depend upon (Chen et al., 2016) external spatial inputs, such as those from place cells in hippocampus. The origin and flow of spatial information within the hippocampal region appears to be quite complicated, and we believe the possibility that grid cells activate place cells cannot be eliminated.

• The model is based on the aperiodic network with feedforward Gaussian envelope function described by Burak and Fiete, 2009, but they also described a periodic solution. The periodic solution has problems with rotation, but some of the results (especially the generation of replays) are affected by or depend on the topology. The authors should clarify the reliance of their results on network topology and whether other formulations may also support their conclusions.

As described in our response to Essential revision1, our new simulations with periodic boundary conditions and uniform drive demonstrate wavefronts that propagate throughout the entire neural sheet.